# Global analysis of p53-regulated transcription identifies its direct targets and unexpected regulatory mechanisms

Mary Ann Allen[1,2,3,4], Zdenek Andrysik[1,2†], Veronica L Dengler[1,2†], Hestia S Mellert[1,2†], Anna Guarnieri[1,2], Justin A Freeman[1,2], Kelly D Sullivan[1,2], Matthew D Galbraith[1,2], Xin Luo[5], W Lee Kraus[5], Robin D Dowell[2,3*], Joaquin M Espinosa[1,2*]

[1]Howard Hughes Medical Institute, University of Colorado, Boulder, Boulder, United States; [2]Department of Molecular, Cellular and Developmental Biology, University of Colorado, Boulder, Boulder, United States; [3]BioFrontiers Institute, Boulder, United States; [4]Computational Biosciences Program, University of Colorado, Denver–Anschutz Medical Campus, Aurora, United States; [5]Signalling and Gene Regulation Laboratory, Cecil H and Ida Green Center for Reproductive Biology Sciences, University of Texas Southwestern Medical Center, Dallas, United States

**Abstract** The p53 transcription factor is a potent suppressor of tumor growth. We report here an analysis of its direct transcriptional program using Global Run-On sequencing (GRO-seq). Shortly after MDM2 inhibition by Nutlin-3, low levels of p53 rapidly activate ~200 genes, most of them not previously established as direct targets. This immediate response involves all canonical p53 effector pathways, including apoptosis. Comparative global analysis of RNA synthesis vs steady state levels revealed that microarray profiling fails to identify low abundance transcripts directly activated by p53. Interestingly, p53 represses a subset of its activation targets before MDM2 inhibition. GRO-seq uncovered a plethora of gene-specific regulatory features affecting key survival and apoptotic genes within the p53 network. p53 regulates hundreds of enhancer-derived RNAs. Strikingly, direct p53 targets harbor pre-activated enhancers highly transcribed in p53 null cells. Altogether, these results enable the study of many uncharacterized p53 target genes and unexpected regulatory mechanisms.

*For correspondence: robin. dowell@colorado.edu (RDD); joaquin.espinosa@colorado.edu (JME)

†These authors contributed equally to this work

## Introduction

The p53 transcription factor is activated by potentially oncogenic stimuli such as ribosomal stress, DNA damage, telomere erosion, nutrient deprivation and oncogene hyperactivation (*Vousden and Prives, 2009*). In the absence of activating signals, p53 is repressed by the oncoproteins MDM2 and MDM4. MDM2 masks the transactivation domain of p53 and is also an E3 ligase that targets p53 for degradation (*Momand et al., 1992*; *Oliner et al., 1993*; *Kubbutat et al., 1997*). MDM4 lacks E3 ligase activity, but represses p53 transactivation potential (*Riemenschneider et al., 1999*). Diverse signaling pathways converge on the p53/MDM2/MDM4 complex to release p53 from its repressors and enable it to regulate transcription of downstream target genes involved in cellular responses such as cell cycle arrest, apoptosis, senescence, autophagy, DNA repair and central metabolism (*Vousden and Prives, 2009*). p53 is inactivated in virtually all human cancers, either by mutations in its DNA binding domain or MDM2/MDM4 overexpression. Significant advances have been made to develop p53-based targeted therapies (*Brown et al., 2009*). One class of small molecules targets the interaction between p53 and its repressors, thus bypassing the need of stress signaling to trigger p53 activation. For example, Nutlin-3, the first-in-class compound, binds to the hydrophobic pocket in MDM2

**eLife digest** The growth, division and eventual death of the cells in the body are processes that are tightly controlled by hundreds of genes working together. If any of these genes are switched on (or off) in the wrong cell or at the wrong time, it can lead to cancer.

It has been known for many years that the protein encoded by one gene in particular—called *p53*—is nearly always switched off in cancer cells. The p53 protein normally acts like a 'brake' to slow the uncontrolled division of cells, and some researchers are working to find ways to switch on this protein in cancer cells. However, this approach appears to only work in specific cases of this disease. For better results, we need to understand how p53 is normally switched on, and what other genes this protein controls once it is activated.

Allen et al. have now identified the genes that are directly switched on when cancer cells are treated with a drug that artificially activates the p53 protein. Nearly 200 genes were switched on, and almost three quarters of these genes had not previously been identified as direct targets of p53.

Although p53 tends to act as a brake to slow cell division, it is not clear how it distinguishes between its target genes—some of which promote cell survival, while others promote cell death. Allen et al. found that survival genes are switched on more strongly than cell death genes via a range of different mechanisms; this may explain why most cancers can survive drug treatments that reactivate p53. Also, Allen et al. revealed that some p53 target genes are primed to be switched on, even before the p53 protein is activated, by proteins (and other molecules) acting in regions of the DNA outside of the genes.

By uncovering many new gene targets for the p53 protein, the findings of Allen et al. could help researchers developing new drugs or treatments for cancer.

required for binding to p53, thus acting as a competitive inhibitor (*Vassilev et al., 2004*). A second class of molecules binds to mutant p53 and partially restores its wild type function (*Brown et al., 2009*). As these compounds enter clinical trials, their efficacy is limited by the fact that p53 activation leads to cancer cell death only in specific scenarios. Thus, there is a clear need to understand how these molecules modulate p53 function and how cell fate choice upon p53 activation is defined. A missing piece in this effort is a definitive elucidation of the direct p53 transcriptome.

Despite its unequivocal importance in cancer biology, our understanding of p53 function as a transcription factor is limited. The protein domains required for DNA binding and transactivation are well characterized, as well as its DNA response elements (p53REs) (*Laptenko and Prives, 2006*). A recent comprehensive survey of the literature identified ~120 genes for which direct regulation has been established (*Riley et al., 2008*), but a comprehensive analysis of p53-regulated RNAs is still missing. Up to this point, the global p53 transcriptional response has been investigated with techniques that measure steady state RNA levels, mostly microarray profiling. These methods require long time points to observe a significant change in the expression of p53-regulated RNAs, which confounds direct vs indirect effects, and additional experiments are required to ascertain direct transcriptional regulation. A popular approach has been to cross-reference microarray data with p53 binding data derived from ChIP-seq assays. Meta-analysis of four recent papers using this strategy indicates that p53 may directly activate >1200 genes, yet only 26 of these genes were commonly activated in all four studies (*Nikulenkov et al., 2012*; *Menendez et al., 2013*; *Schlereth et al., 2013*; *Wang et al., 2013*) (see later, *Figure 2—figure supplement 1*). It is unclear to what extent this lack of overlap is due to methodological differences and/or cell type-specific differences in direct p53 action vs post-transcriptional regulation.

We report here the first genome-wide analysis of p53-regulated RNA synthesis. Using Global Run-On sequencing (GRO-seq) (*Core et al., 2008*), we ascertained direct regulation by using a short time point of Nutlin-3 treatment in isogenic cell lines with or without p53. Strikingly, Nutlin leads to p53-dependent transcriptional activation of hundreds of genomic loci prior to any significant increase in total p53 levels, thus highlighting the critical role of MDM2 in masking the p53 transactivation domain. Comparative global analysis of RNA synthesis by GRO-seq vs RNA steady state levels by microarray revealed that many p53 target genes transcribed at low levels are missed by microarray

experiments. Strikingly, p53 represses the basal expression of a subset of its target genes before MDM2 inhibition. GRO-seq uncovered many gene-specific transcriptional events affecting key survival and apoptotic genes within the network, including the occurrence of bidirectional promoters (both overlapping and non-overlapping), clustered transactivation and various forms of antisense transcription. GRO-seq revealed widespread activation of enhancer-derived RNAs (eRNAs) arising from p53REs. Interestingly, direct p53 target genes harbor 'pre-activated' p53REs, as defined by the strong production of eRNAs in the isogenic p53 null cells. These results elucidate novel p53-regulated RNAs as well as gene-specific regulatory events within the p53 network and pave the road for a myriad of future mechanistic studies.

## Results

### p53 rapidly activates a multifunctional transcriptional program upon MDM2 inhibition

In order to study the direct transcriptional response upon p53 activation, we performed GRO-seq in isogenic HCT116 p53 +/+ and p53 −/− cell lines. After several hours of treatment with 10 μM Nutlin-3a (referred hereto as Nutlin), only p53 +/+ cells undergo cell cycle arrest and display induction of many known p53 target genes (*Tovar et al., 2006*; *Henry et al., 2012*). Using GRO-seq, we investigated the effects of Nutlin vs vehicle (DMSO) in both cell lines under conditions of exponential cell proliferation, when p53 levels are low and cell cultures display virtually no signs of cell cycle arrest or apoptosis. To minimize the possibility of measuring indirect effects, we chose a 1 hr time point of Nutlin treatment. Typical GRO-seq results are shown for two well-characterized p53 target genes, *CDKN1A* (*p21*) and *TP53I3* (*PIG3*) (*el-Deiry et al., 1993*; *Polyak et al., 1997*), which display significant increase in GRO-seq signals upon Nutlin treatment only in HCT116 p53 +/+ cells (*Figure 1A*, *Figure 1—figure supplement 1A*, respectively). Not surprisingly, Q-RT-PCR demonstrates that the steady state expression of the CDKN1A and TP53I3 mRNAs does not increase at the 1 hr time point as used for GRO-seq (*Figure 1B*). In fact, a significant increase in the steady state levels of both mRNAs requires several hours of p53 activation. Furthermore, Western blot analysis shows that 1 hr of Nutlin treatment does not increase total p53 or p21 protein levels to a significant degree (*Figure 1C*). Analysis of cell cycle progression using BrdU incorporation assays shows no signs of cell cycle arrest at the 1 hr time point, but a clear reduction in S phase cells is evident at 12 hr only in the p53 +/+ cells (*Figure 1C*, *Figure 1—figure supplement 1B*). Overall, these observations indicate that our GRO-seq analysis using a 1 hr time point of p53 activation would largely preclude secondary effects driven by well-established direct p53 target genes, such as *CDKN1A* and *TP53I3*, thus enabling the identification of the direct p53 transcriptome.

Next, we used the DESeq algorithm (*Anders and Huber, 2010*) to identify annotated gene loci whose transcription is significantly changed upon Nutlin treatment (see 'Materials and methods' for details). Using a cut-off of adjusted $p(a) < 0.1$, we identified 198 gene loci whose transcription is significantly induced upon Nutlin treatment in p53 +/+ cells (*Figure 1D*; *Supplementary file 1*). This analysis identified only four gene loci whose transcription was diminished in the p53 +/+ cells (*FLVCR2*, *NR4A3*, *RELB* and *EGR1*); however, none of these genes showed reductions in steady state mRNA levels upon prolonged p53 activation (see later, *Figure 2*). The specificity of Nutlin is demonstrated by the negligible changes observed in p53 −/− cells, where our analysis identified 5 induced and 2 repressed genes, all of which have less than 1.5-fold changes and none of which was among those differentially transcribed in p53 +/+ cells (*Figure 1D*). From this point forth, we focused on the 198 genes activated in the p53 +/+ cells, which we considered to be the direct p53 transcriptional program in this cell type. The notion that these genes are indeed direct p53 targets is reinforced by the observation that most of them (176 out of 198) show an increase in transcription as early as 30 min after Nutlin addition to the cell culture (*Figure 1—figure supplement 1C*). Of these 198 genes, 55 were known validated direct p53 targets, 66 were targets predicted by one or more published microarray/ChIP-seq studies, and 77 are putative novel direct p53 targets (*Figure 1—figure supplement 1D*, a comprehensive annotation of these genes is provided in *Supplementary file 1*). Q-RT-PCR validation showed that novel genes are induced at a 12 hr time point of Nutlin treatment at the mRNA steady state level to a degree comparable to those genes predicted by published microarray/ChIP-seq studies (*Figure 1E*). Furthermore, 12 out of the 14 novel p53 target genes tested are also induced at the mRNA steady state level when using doxorubicin, a DNA-damaging agent that activates p53 via stress

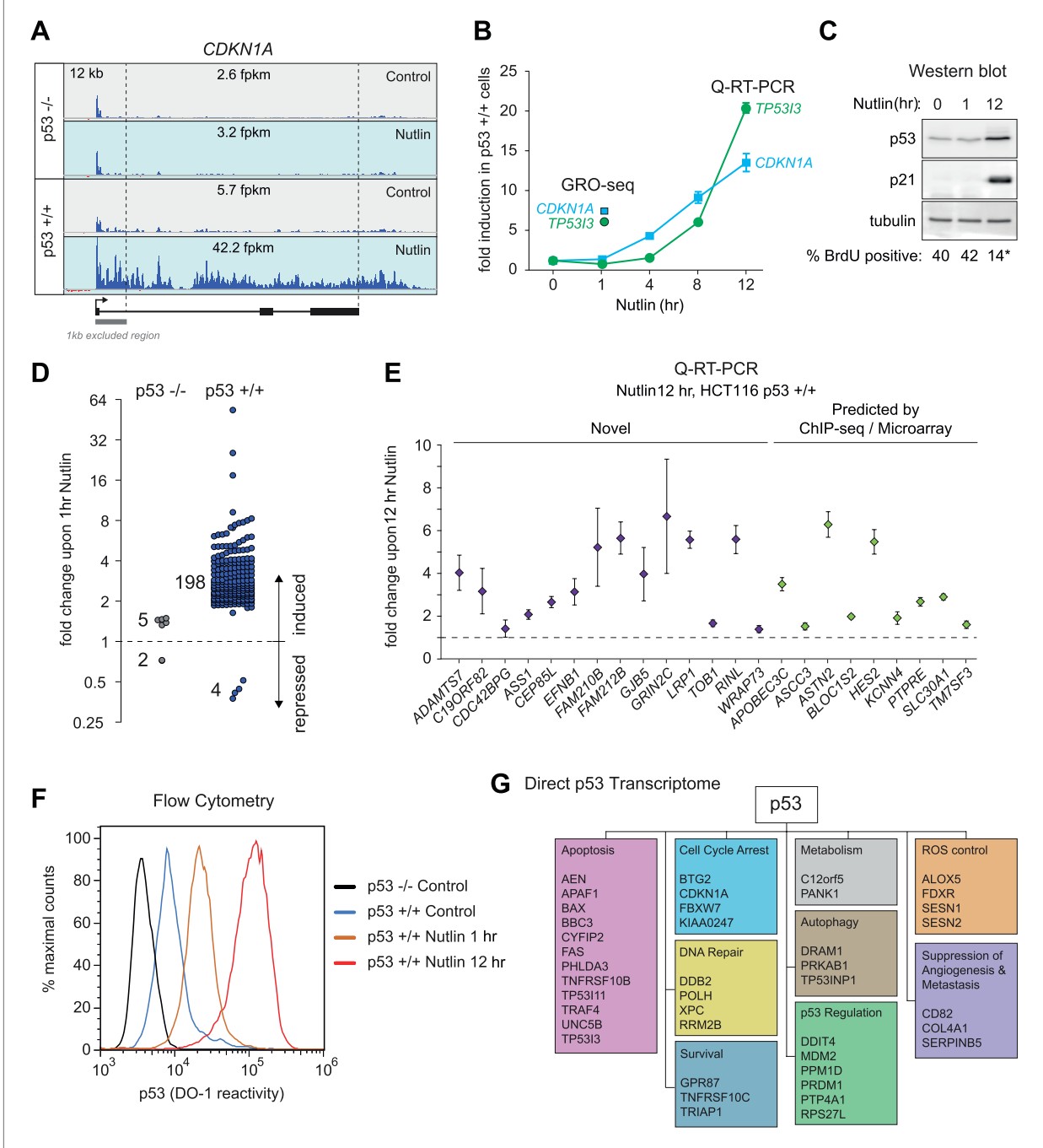

**Figure 1**. GRO-seq analysis of the p53 transcriptional program. (**A**) GRO-seq results for the p53 target locus *CDKN1A* (*p21*). Isogenic p53 −/− and p53 +/+ HCT116 cells were treated for 1 hr with either 10 µM Nutlin-3a (Nutlin) or vehicle (DMSO, Control). Fragments per kilobase per million reads (fpkm) are shown for the intragenic region. The first kilobase downstream of the transcription start site (TSS) was excluded from the fpkm calculation to minimize effects of RNAPII pausing. The total genomic region displayed is indicated in the top left corner. Blue signals are reads mapping to the sense strand, red signals are reads mapping to the antisense strand. See *Figure 1—figure supplement 1A* for results of the *TP53I3* locus. (**B**) GRO-seq detects transactivation of the canonical p53 target genes *CDKN1A* and *TP53I3* at 1 hr of Nutlin treatment, prior to any detectable increase in steady state mRNA levels as measured by Q-RT-PCR. (**C**) A 1 hr time point of Nutlin treatment does not produce significant p53 accumulation, p21 protein induction or a decrease in number of S phase cells as measured by BrdU incorporation assays. * indicates p<0.05. See also *Figure 1—figure supplement 1B* for quantification data of BrdU assays. (**D**) Genome-wide analysis using the DESeq algorithm identifies 198 annotated gene loci transactivated upon Nutlin treatment only in HCT116 p53 +/+ cells. See *Supplementary file 1* for a detailed annotation of these genes. (**E**) Q-RT-PCR validates induction of novel and predicted direct p53 target genes upon 12 hr of Nutlin treatment. mRNA expression

*Figure 1. Continued on next page*

*Figure 1. Continued*

was normalized to 18s rRNA values and expressed as fold change Nutlin/DMSO. Data shown are the average of three biological replicates with standard errors from the mean. (**F**) Flow cytometry analysis using the DO-1 antibody recognizing the MDM2-binding surface in the p53 transcactivation domain 1 (TAD1) reveals increased reactivity as early as 1 hr of Nutlin treatment, indicative of unmasking of the TAD1 at this early time point. (**G**) p53 directly activates a multifunctional transcriptional program at 1 hour of Nutlin treatment, including many canonical apoptotic genes. See *Supplementary file 1* for a complete list and annotation.

The following figure supplements are available for figure 1:

**Figure supplement 1**. GRO-seq reveals the immediate direct p53 transcriptional response.

signaling cascades (*Lowe et al., 1994*), thus revealing that transactivation of most novel genes is not unique to pharmacological inhibition of MDM2 (*Figure 1—figure supplement 1E*). Finally, we investigated whether activation of novel p53 targets can also be observed at the protein level. Indeed, Western blot analysis demonstrates protein induction for the novel genes GRIN2C, PTCDH4 and RINL (*Figure 1—figure supplement 1F*). Thus, our GRO-seq experiment clearly expands the universe of direct p53 target genes, paving the road for mechanistic studies investigating the function of these genes in the p53 network.

Although it is known that MDM2 represses p53 by both masking its transactivation domain and also targeting it for degradation (*Momand et al., 1992*; *Oliner et al., 1993*; *Kubbutat et al., 1997*), it has been difficult to dissect to what extent each mechanism contributes to repression of p53 target genes in diverse functional categories. Studies employing steady state mRNA measurements concluded that prolonged p53 activation and/or higher levels of cellular p53 were required for activation of apoptotic genes, some of which display delayed kinetics of induction at the mRNA steady state level as compared to cell cycle arrest genes (*Chen et al., 1996*; *Zhao et al., 2000*; *Szak et al., 2001*; *Espinosa et al., 2003*; *Das et al., 2007*). However, GRO-seq demonstrates that a 1 hr time point of Nutlin treatment induces transcription of genes in every major pathway downstream of p53 (*Supplementary file 1*). The observation that key survival and apoptotic genes (e.g., *CDKN1A*, *TP53I3*) show greater than sixfold increase in transcription at a time point preceding a proportional increase in total p53 levels (*Figure 1A,C*, *Figure 1—figure supplement 1A*), suggests that the mere unmasking of the p53 transactivation domain suffices to activate a multifaceted transcriptional program. To further test this notion, we performed flow cytometry analyses using a monoclonal antibody (DO-1) that recognizes an epitope in the p53 N-terminal transactivation domain 1 (TAD1) that overlaps with the MDM2-binding surface competed by Nutlin (*Picksley et al., 1994*). In fact, the DO-1 antibody competes the p53-MDM2 interaction in vitro in analogous fashion to Nutlin (*Cohen et al., 1998*). Under the denaturing conditions of a Western Blot assay, where p53-MDM2 complexes are fully disrupted, this antibody shows no significant increase in total p53 levels at the 1 hr time point of Nutlin treatment (*Figure 1C*). However, we posited that DO-1 reactivity should increase significantly upon Nutlin treatment under the fixed conditions employed in flow cytometry. Expectedly, flow cytometry quantitation shows that, even before Nutlin treatment, p53 +/+ cells have significantly more DO-1 reactivity than p53 −/− cells (*Figure 1F*). The functional importance of this 'basal p53 activity' will be investigated later in this report (*Figure 3*). Interestingly, the p53 +/+ cell population shifts to significantly higher DO-1 reactivity at the 1 hr time point, as predicted by epitope unmasking. A further increase is observed at 12 hr of Nutlin treatment, when total p53 levels have risen considerably as measured by Western blots (*Figure 1C,F*). Finally, since GRO-seq is a population average experiment, we performed immunofluorescence assays to test if our GRO-seq results could be explained by massive p53 accumulation in just a few outlier cells within the population at the 1 hr time point. However, these experiments discarded the notion of outlier cells: although ~3% cells show high p53 staining at the 1 hr time point, this number is not significantly different than observed in control p53 +/+ cells (*Figure 1—figure supplement 1G,H*).

Altogether, these results indicate that the low levels of p53 present in proliferating cancer cells suffice to directly activate a multifunctional transcriptional program, including many canonical apoptotic genes, upon unmasking of the p53 transactivation domain by Nutlin. However, as discussed later in the paper (*Figure 4*), this conclusion does not necessarily conflict with previous reports showing

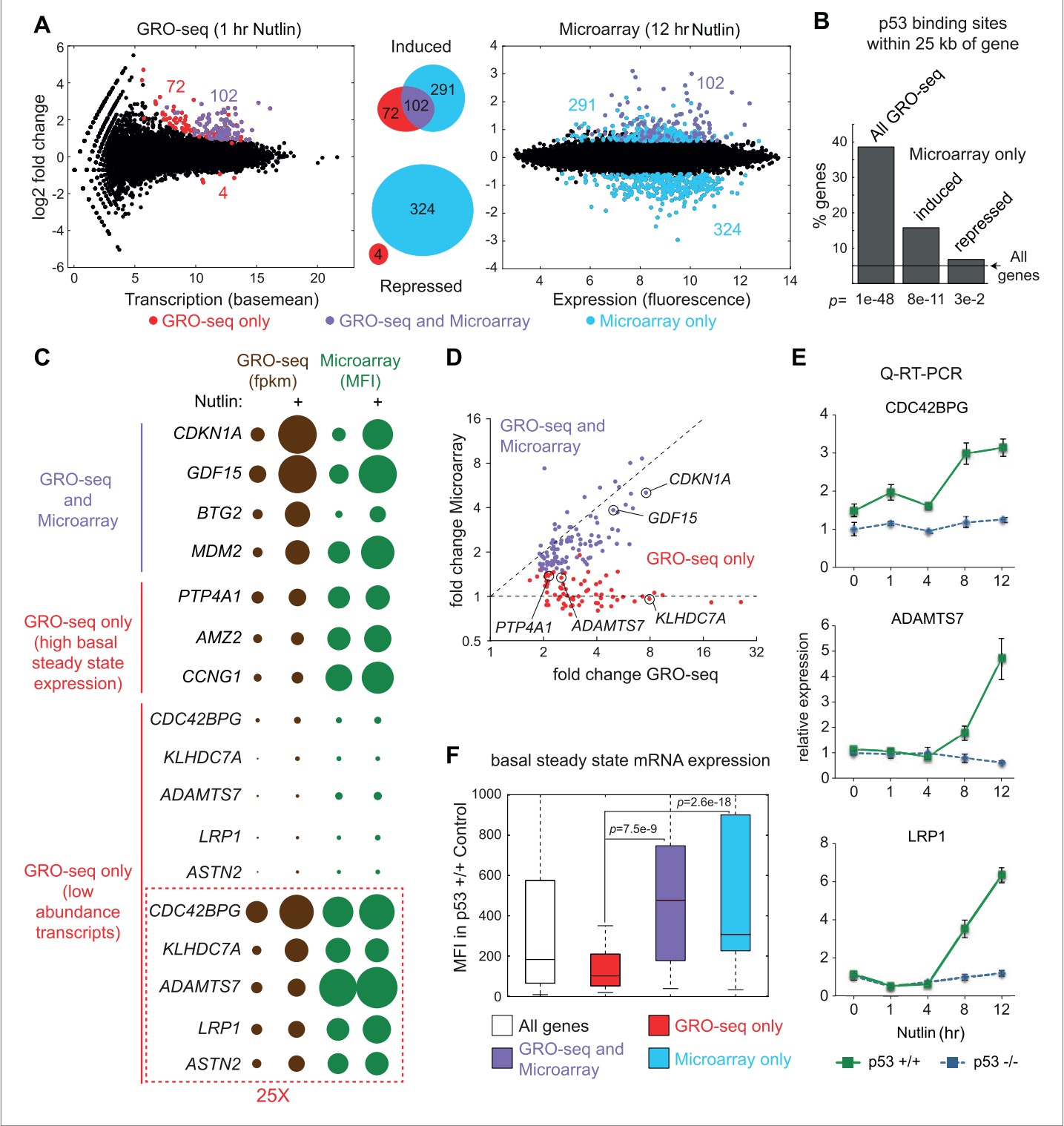

**Figure 2**. Global analysis of p53 effects on RNA synthesis vs steady state levels. (**A**) MAplots for GRO-seq and microarray gene profiling experiments in HCT116 p53 +/+ cells after 1 hr and 12 hr of Nutlin treatment, respectively. Colors indicate whether genes scored as statistically different in both platforms (purple), in the GRO-seq only (red) or the microarray experiment only (blue). (**B**) Few genes downregulated in the microarray experiment show p53 binding within 25 kb of the gene, suggestive of indirect regulation. (**C**) Bubble plots displaying relative signals derived from the GRO-seq and microarray experiments illustrate how genes with very high basal expression or very low transcription are not significantly affected at the steady state level as measured by microarray. For the *CDC42BPG*, *KLHDC7A*, *ADAMTS7*, *LRP1* and *ASTN2* loci,

*Figure 2. Continued on next page*

*Figure 2. Continued*

the signals were replotted at 25-fold magnification. (**D**) Scatter plot showing comparative fold induction for p53 target genes transactivated at 1 hr Nutlin treatment between the GRO-seq and microarray experiments. (**E**) Q-RT-PCR indicates that many low abundance transcripts upregulated by GRO-seq are indeed induced at the steady state level. (**F**) Box and whisker plots showing the expression of various gene sets as detected by microarray.

The following figure supplements are available for figure 2:

**Figure supplement 1**. Mechanisms of indirect gene repression by p53.

**Figure supplement 2**. ChIP analysis of novel p53 target genes.

differential timing of mRNA accumulation between arrest and apoptotic genes as seen by steady state RNA measurements.

## Global analysis of p53 effects on RNA synthesis vs RNA steady state levels

The global p53 transcriptional response has been previously investigated using measurements of RNA steady state levels (i.e., microarray profiling) and p53 chromatin binding (e.g., ChIP-seq). Meta-analysis of four recent reports using this approach indicates that >1200 genes are putative direct targets of p53 transactivation, yet only 26 are common between the four studies (*Figure 2— figure supplement 1A,B*; *Supplementary file 2*) (*Nikulenkov et al., 2012*; *Menendez et al., 2013*; *Schlereth et al., 2013*; *Wang et al., 2013*). Furthermore, these studies suggest 80 genes that could be directly repressed by p53, yet none are shared between any two studies (*Figure 2— figure supplement 1A,B*; *Supplementary file 2*). In order to investigate how GRO-seq analysis of the immediate p53 transcriptional response would compare to a global analysis of RNA steady state levels, we performed a microarray analysis of HCT116 p53 +/+ cells after 12 hr of Nutlin treatment, a time point similar to that used in the previous studies. Several important observations arise from this comparison.

First, there is a clear lack of overlap between the two analyses (*Figure 2A*). Among the induced genes identified by the two experimental platforms, only 102 are common. 291 genes are called as induced by the microarray experiment only. This group would include genes whose transcription may be stimulated at later time points via indirect mechanisms, but may also include true direct p53 target genes that require higher levels of p53 to be activated. For example, we noted that the canonical p53 target gene *GADD45A* fell in this group, as its transcription was mildly induced at 1 hr and thus fell below our statistical cut-off. Interestingly, 72 genes were identified as induced by GRO-seq only, despite the fact that the microarrays utilized harbored multiple probes against these mRNAs. The possible explanations for this finding are discussed below.

Second, microarrays detect 324 genes repressed upon 12 hr of Nutlin treatment, none of which were called as repressed by GRO-seq. The mechanism of p53-mediated gene repression remains debated in the field. Multiple independent ChIP-seq studies concur in that p53 binds weakly and very distally to those gene loci whose mRNAs are downregulated at the steady state level, and that the p53REs found at these sites match poorly to the consensus DNA sequence (*Nikulenkov et al., 2012*; *Menendez et al., 2013*; *Schlereth et al., 2013*; *Wang et al., 2013*). Using seven different available global ChIP datasets derived from HCT116 and two other cell lines, we created a collection of high confidence p53 binding events to analyze p53 binding in the vicinity of the various gene groups ('Materials and methods'). Nearly 40% of the 198 genes induced by GRO-seq harbor a p53 binding event within 25 kb, significantly more than expected from random occurrence ($p=1e-48$, Hypergeometric test) (*Figure 2B*). Among the genes induced by microarray only, nearly 15% harbored p53 binding within 25 kb, still significantly more than expected by chance ($p=8e-11$), which suggests that some of these genes may be true direct targets activated at later time points. Most importantly, genes considered as repressed by the microarray profiling show little p53 binding within 25 kb, barely above what is expected by chance ($p=3e-2$), suggesting that the repression observed is largely indirect.

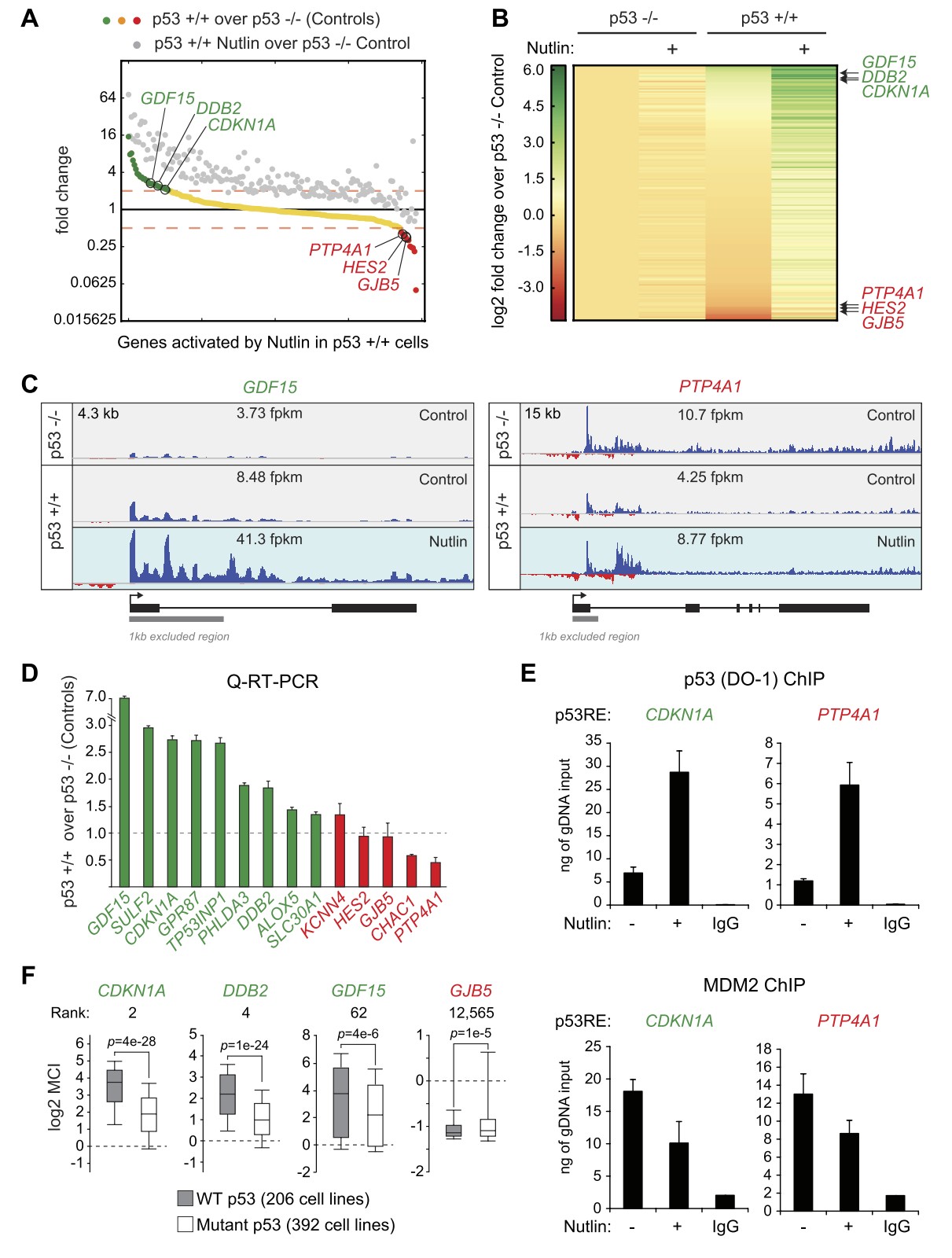

**Figure 3**. p53 exerts varying activating and repressing effects on its target genes prior to MDM2 inhibition. (**A**) 198 genes activated upon 1 hr Nutlin treatment in HCT116 p53 +/+ cells are ranked from left to right based on their basal transcription in p53 +/+ cells over p53 −/− cells. Green indicates genes whose basal transcription is greater than twofold in p53 +/+ cells, red indicates lesser than twofold. Grey dots display the transcription of the

*Figure 3. Continued on next page*

*Figure 3. Continued*

same genes in Nutlin-treated p53 +/+ cells. (**B**) Heatmap displaying relative transcriptional activity of direct p53 target genes identified by GRO-seq relative to control p53 −/− cells. Genes are sorted based on their transcription in control p53 +/+ cells. (**C**) Genome browser views of representative genes whose basal transcription is higher (*GDF15*) or lower (*PTP4A1*) in the presence of MDM2-bound p53. See *Figure 3—figure supplement 1A* for matching RNAPII ChIP data. (**D**) Q-RT-PCR measurements of genes whose basal transcription was found to be 2x higher (green) or lower (red) in the presence of MDM2-bound p53. (**E**) ChIP assays show binding of p53 and MDM2 to the p53REs in the *CDKN1A* and *PTP4A1* gene loci (−2283 bp and +1789 relative to TSS, respectively), prior to inhibition of the p53-MDM2 interaction by Nutlin. Nutlin treatment leads to increased p53 signals with the DO-1 antibody recognizing the p53 TAD1, concurrently with a decrease in MDM2 signals. MDM2 ChIP was performed in SJSA cells carrying a MDM2 gene amplification **F**. Oncomine gene expression analysis of 598 cancer cell lines of varied p53 status shows that *CDKN1, DDB2* and *GDF15* are more highly expressed in wild type p53 cell lines, whereas *GJB5* is more highly expressed in mutant p53 cell lines. The ranking position of these genes is also indicated.

The following figure supplements are available for figure 3:

**Figure supplement 1**. Differential effects of p53 on the basal transcription of its target genes.

**Figure supplement 2**. p53 mutational status affects the basal expression of its target genes.

Indirect gene repression downstream of p53 activation could be mediated at the post-transcriptional level by p53-inducible miRNAs, and/or at the transcriptional level by the action of direct p53 targets known to repress transcription. Of note, GRO-seq identified 5 miRNAs directly transactivated by p53 (miR-1204, miR-3189, miR-34a, miR4679-1 and miR-4692, see *Supplementary file 1*). Most prominent among these is miR-34a, a well characterized p53-inducible miRNA known to mediate indirect repression by p53 at late time points. In fact, we found that nearly 72% of genes repressed in our microarray by Nutlin were previously shown by others (*Lal et al., 2011*) to be downregulated upon overexpression of miR-34a in HCT116 cells (p<2.2e−16, Hypergeometric test, *Figure 2—figure supplement 1C*). A recent report demonstrated that p21 and E2F4, a transcriptional repressor of S-phase genes acting coordinately with co-repressors of the RB family, are required for the downregulation of many genes previously characterized as 'direct' targets of p53 repression (*Benson et al., 2013*). In agreement with these published findings, Ingenuity Pathway Analysis (IPA) of the genes repressed in our microarray experiment revealed that the top three regulators affecting these genes are indeed E2F4 (p=1.02e−81, Fisher's Exact Test), CDKN1A (p=8.21e−62) and RB (p=8.12e−60) (*Figure 2—figure supplement 1D*). Altogether, these data indicate that most gene repression observed in our system is likely to be indirect, either via miRNAs, such as miR-34a, and/or the p21>RB>E2F4 axis (*Figure 2—figure supplement 1E*). Of note, GRO-seq also identified the transcriptional repressor PRDM1 (BLIMP1) as a direct target of p53 (*Yan et al., 2007*), revealing yet another possible mechanism for indirect gene repression downstream of p53. However, IPA did not identify PRDM1 as a top regulator of genes repressed in the microarray study (not shown).

Given that 72 genes were identified as activated only by GRO-seq, we further investigated the possible reasons for this result. Analysis of absolute signals generated by the GRO-seq (fpkm) vs microarray (mean fluorescence intensity, MFI) experiments generated several important insights (*Figure 2C*). First of all, robust p53 target genes such as *CDKN1A* and *GDF15* show strong increases in both platforms. When these results are expressed as fold induction, a strong correlation between transcriptional output and steady state levels is evident for these genes (*Figure 2D*). However, transcriptional output often does not correlate with steady state mRNA levels. For example, while the *BTG2* locus has a greater transcriptional output upon p53 activation than the *MDM2, PTP4A1, AMZ2* and *CCNG1* loci, its steady state mRNA levels are much lower, likely due to post-transcriptional repression of *BTG2* (*Figure 2C*). This would explain why a small group of 'GRO-seq only' genes, including known p53 targets such as *PTP4A1* and *CCNG1*, are not called by the microarray experiments: they display very high basal steady state levels, which do not increase significantly despite clear transcriptional induction (*Figure 2C*). However, most 'GRO-seq only' genes belong to a different category, as marked by their very low levels of transcription and steady state mRNA expression. Genes like *CDC42BPG, ADAMTS7* and *LRP1* are clearly induced at the transcriptional level but show no apparent increase in the microarray signals (*Figure 2C,D*). Remarkably, the induction of these mRNAs upon p53 activation is evident by Q-RT-PCR at the time point of the microarray experiment (*Figure 2E*). In fact, when microarray-derived signals are displayed for the various gene sets, the 'GRO-seq only' group shows very significant lower expression as compared to all other groups (*Figure 2F*). Altogether, we

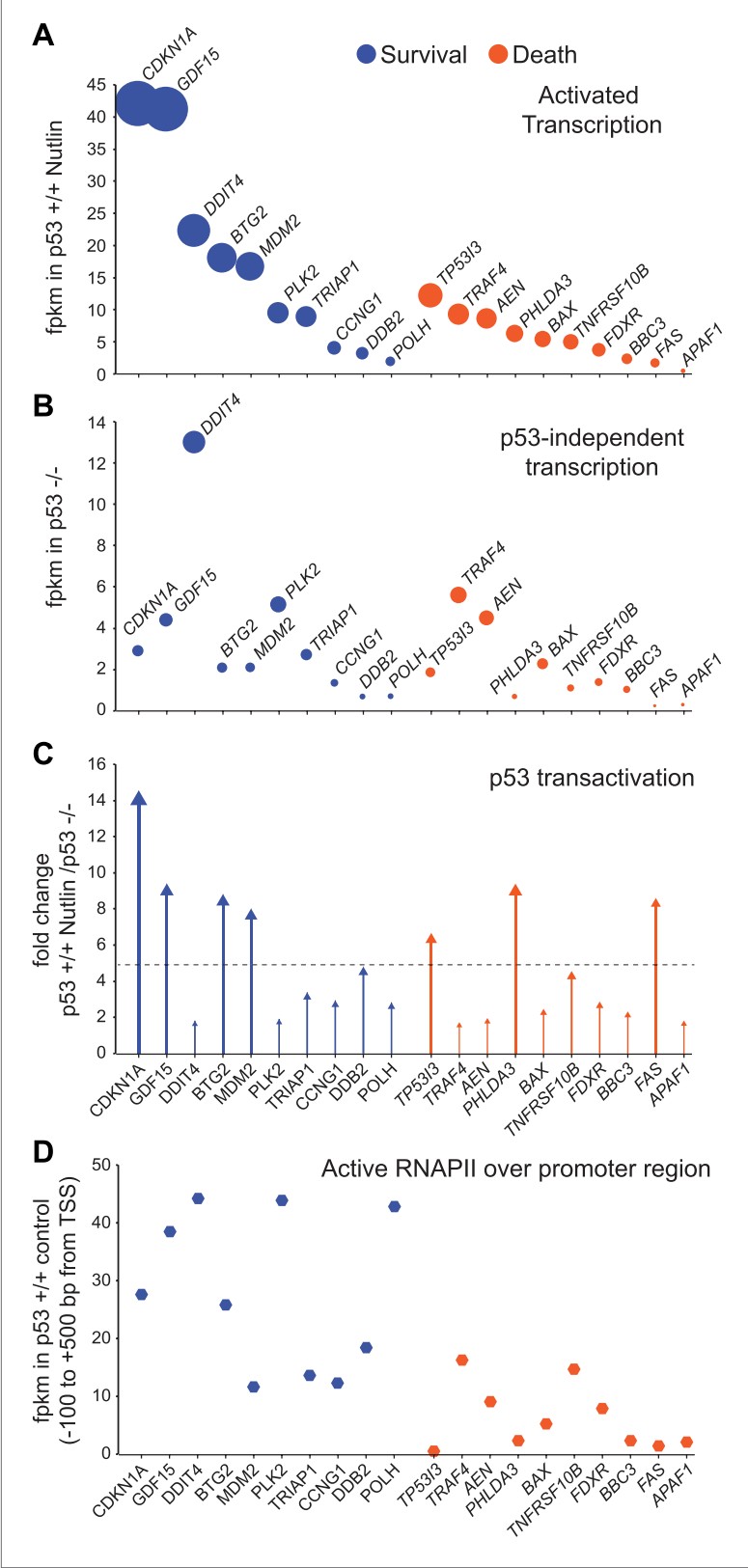

**Figure 4**. GRO-seq analysis of key survival and death genes within the p53 network. (**A**) The 10 most transcribed pro-survival and pro-apoptotic genes identified by GRO-seq ranked by decreasing transcriptional output in Nutlin-treated p53 +/+ cells. The surface of the bubbles represents the GRO-seq signal output relative to the
*Figure 4. Continued on next page*

*Figure 4. Continued*

*CDKN1A* locus. (**B**) Transcriptional output of same genes shown in **A** in p53 −/− cells. (**C**) Fold change analysis showing the overall effect of p53 on the transcription of its survival and apoptotic targets. (**D**) Survival genes within the p53 network tend to carry more proximally bound, transcriptionally engaged RNAPII over their promoter regions than apoptotic genes.
The following figure supplements are available for figure 4:

**Figure supplement 1**. p53 target genes display a wide range of RNAPII pausing and promoter divergence.
**Figure supplement 2**. Examples of gene-specific features affecting key pro-apoptotic and survival p53 target genes.

conclude that microarray profiling is not sensitive enough to detect these low abundance transcripts, which could explain why several published ChIP-seq/microarray studies failed to identify these genes as direct p53 targets. Alternatively, it is possible that p53 binds to these genes from very distal sites outside of the arbitrary window defined during bioinformatics analysis of ChIP-seq data. To discern among these possibilities, we analyzed ChIP-seq data in search of high confidence p53 binding events in the vicinity of several novel genes identified by GRO-seq, and evaluated p53 binding using standard ChIP assays. Indeed, we detected clear p53 binding to all p53REs tested at these novel p53 targets (*Figure 2—figure supplement 2*). Of note, p53 binds to proximal regions at the *CDC42BPG* and *LRP1* loci (+1373 bp and −694 bp relative to transcription start site [TSS], respectively), indicating that these genes could have been missed in previous studies due to the low abundance of their transcripts. In contrast, p53 binds to very distal sites (i.e., >30 kb from the TSS) at the *ADAMTS7*, *TOB1*, *ASS1* and *CEP85L* loci (*Figure 2—figure supplement 2*), suggesting that these genes would have been missed as direct targets when setting an arbitrary <30 kb window during ChIP-seq analysis. In summary, GRO-seq enables the identification of novel direct p53 target genes due both to its increased sensitivity and the fact that it does not require proximal p53 binding to ascertain direct regulation.

## p53 represses a subset of its direct target genes prior to MDM2 inhibition

Others and we have observed that in proliferating cells with minimal p53 activity, p53 increases the basal expression of some of its target genes (*Tang et al., 1998*; *Espinosa et al., 2003*). This was first recorded for *CDKN1A* (*Tang et al., 1998*), and it is confirmed by our GRO-seq analysis (*Figure 1A*, compare 2.6 to 5.7 fpkm in the Control tracks). To investigate whether this is a general phenomenon we analyzed the basal transcription of all p53-activated genes in control p53 +/+ vs p53 −/− cells (*Figure 3A,B*). Interestingly, p53 status exerts differential effects among its target genes prior to MDM2 inhibition with Nutlin. While many genes show the same behavior as *CDKN1A* (e.g., *GDF15*, *DDB2*, labeled green throughout *Figure 3*), another group shows decreased transcription in the presence of MDM2-bound p53 (e.g., *PTP4A1*, *HES2*, *GJB5,* labeled red throughout *Figure 3*). Genome browser views illustrating this phenomena are provided for *GDF15* and *PTP4A1* in *Figure 3C*. The differential behavior of RNAPII at these gene loci is also observed in ChIP assays using antibodies against the Serine 5- and Serine 2-phosphorylated forms of the RBP1 C-terminal domain repeats, which mark initiating and elongating RNAPII complexes, respectively (S5P- and S2P-RNAPII, *Figure 3—figure supplement 1A*). Whereas the 'basally activated' *GDF15* locus displays higher GRO-seq and RNAPII ChIP signals in untreated p53 +/+ cells, the 'basally repressed' *PTP4A1* locus shows lower signals in the presence of MDM2-p53 complexes. Such differential effects among p53 target genes have a clear impact on their absolute level of transcription upon MDM2 inhibition: whereas those whose basal activity was increased ranked among the most differentially transcribed between Nutlin-treated p53 +/+ and p53 −/− cells (top of the heatmap in *Figure 3B*); those basally repressed are virtually no different in expression between Nutlin-treated p53 +/+ and p53 −/− cells (bottom of the heatmap in *Figure 3B*). Importantly, Q-RT-PCR shows that the differential effects of p53 on the basal transcription of its targets are generally translated into differences in mRNA steady level (*Figure 3D*).

Overall, these results indicate that p53 acts as a repressor at a subset of its targets in a manner that is relieved by Nutlin, suggesting that MDM2 recruitment by basal levels of p53 may repress transcription at specific loci. To test this hypothesis, we performed ChIP experiments for p53 and MDM2 under

conditions matching the GRO-seq experiment. For the p53 ChIP, we employed the monoclonal antibody DO-1 that recognizes the p53 TAD1 and whose reactivity should increase upon displacement of MDM2 by Nutlin. Importantly, ChIP assays show a significant amount of chromatin-bound p53 above background levels at the p53REs in the *CDKN1A* locus (basally activated), and the *PTP4A1* and *HES2* loci (basally repressed), even before Nutlin treatment (*Figure 3E*, *Figure 3—figure supplement 1B*). Of note, the DO-1 ChIP signals increase upon Nutlin treatment, as expected from epitope unmasking. ChIP assays also detect MDM2 chromatin binding above background levels at these three p53REs, with signals decreasing upon Nutlin treatment, as expected by the competitive action of this molecule (*Figure 3E*, *Figure 3—figure supplement 1B*). Of note, although Nutlin disrupts the interaction between the p53 N-terminus and the hydrophobic pocket in the N-terminal domain of MDM2, a second molecular interaction occurs between the p53 C-terminus and the MDM2 N-terminus that is not competed by Nutlin in vitro (*Poyurovsky et al., 2010*), which may explain why the MDM2 signal is not completely abrogated upon a short time point of Nutlin treatment. Western blots demonstrating specific MDM2 immunoprecipitation under the ChIP conditions utilized are shown in *Figure 3—figure supplement 1C*.

Next, we investigated whether the differential effects of basal p53 levels on expression of its direct targets could be revealed in an analysis of hundreds of cell lines expressing wild type vs mutant p53. More specifically, we hypothesized that genes that are basally transactivated by p53 would be more highly expressed in p53 WT cells than transrepressed genes. Oncomine analysis of 598 cancer cell lines not treated with p53-activating agents revealed that many genes that are 'basally activated' in HCT116 cells such as *CDKN1A*, *DDB2* and *GDF15* indeed show significantly higher mRNA expression in WT p53 cell lines (*Figure 3F*). In contrast, the 'basally repressed' gene *GJB5* shows significantly higher expression in mutant p53 cell lines. When all 12,624 genes in the Oncomine analysis are ranked according to their relative expression in WT p53 over mutant p53 cell lines, many genes whose basal transcription is upregulated by p53 in HCT116 cells appear at the top of this ranking (e.g., *CDKN1A*, *DDB2* and *GDF15*, ranked 2, 4 and 62, respectively) (*Figure 3—figure supplement 2A*). However, some direct targets 'basally repressed' by p53, such as *GJB5*, show an inverse correlation with WT p53 status. Collectivelly, the direct p53 targets identified by GRO-seq are enriched toward the top of the ranking, which is revealed in a Gene set enrichment analysis (GSEA) (*Figure 3—figure supplement 2A*). In contrast, genes induced only in the microarray platform (i.e., mostly indirect targets) do not show strong enrichment in a GSEA analysis. When the relative basal transcription between HCT116 p53 +/+ and p53 −/− cells is plotted against the relative mRNA expression in p53 WT vs p53 mutant cell lines, it is apparent that many 'basally activated' genes are more highly expressed in p53 WT cells (green dots in the upper right quadrant in *Figure 3—figure supplement 2B*). Finally, the differential pattern of basal expression among p53 targets is also observed in an analysis of 256 breast tumors for which p53 status was determined, where *CDKN1A*, *DDB2* and *GDF15* (but not *GJB5*) show higher expression in the p53 WT tumors (*Figure 3—figure supplement 2C*).

Altogether, these results reveal a qualitative difference among p53 target genes in terms of their sensitivity to basal p53-MDM2 complexes as depicted in *Figure 3—figure supplement 2D*. Although indirect effects can not be fully ruled out, the fact that we can detect p53 and MDM2 binding to the p53REs near these gene loci suggest direct action. Of note, early in vitro transcription studies demonstrated that MDM2 represses transcription when tethered to DNA independently of p53, which may provide the molecular mechanism behind our observations (*Thut et al., 1997*) ('Discussion').

## GRO-seq reveals gene-specific regulatory mechanisms affecting key survival and apoptotic genes

Many research efforts have been devoted to the characterization of molecular mechanisms conferring gene-specific regulation within the p53 network, as these mechanisms could be exploited to manipulate the cellular response to p53 activation. Most research has focused on factors that differentially modulate p53 binding or transactivation of survival vs apoptotic genes (*Vousden and Prives, 2009*). GRO-seq identified a plethora of gene-specific regulatory features affecting p53 targets, but our analysis failed to reveal a universal discriminator between survival and death genes within the network.

When direct p53 target genes with well-established pro-survival (i.e., cell cycle arrest, survival, DNA repair and negative regulation of p53) and pro-death (i.e., extrinsic and intrinsic apoptotic signaling) functions are ranked based on their transcriptional output in Nutlin-treated p53 +/+ cells, it is evident that key pro-survival genes such as *CDKN1A*, *GDF15*, *BTG2* and *MDM2* are transcribed at much

higher rates than any apoptotic gene (*Figure 4A*). For example, ~20-fold more RNA is produced from the *CDKN1A* locus than from the *BBC3* locus encoding the BH3-only protein PUMA. The most potently transcribed apoptotic gene is *TP53I3* (*PIG3*), yet its transcriptional output is still 3.4-fold lower than *CDKN1A*. Based on measurements of steady state RNA levels, it was observed that apoptotic genes such as *TP53I3* and *FAS* are induced with a slower kinetics than *CDKN1A* (*Szak et al., 2001*; *Espinosa et al., 2003*). However, GRO-seq suggests that this is not due to a slower kinetics of RNAPII transactivation, but rather to a lower transcriptional output from the apoptotic loci. Although both *CDKN1A* and *TP53I3* display similar transcriptional induction within 1 hr of Nutlin treatment (7.4 and 6.07 fold induction, *Supplementary file 1*), the *TP53I3* mRNA takes longer to display increased accumulation over basal levels (see Q-RT-PCR in *Figure 1B*) (*Szak et al., 2001*). Thus, differences in mRNA induction kinetics between gene classes could be explained from differential amounts of RNA synthesis. To investigate how much of the differential transcriptional output is due to p53 action vs p53-autonomous mechanisms, we analyzed the activity of these gene loci in p53 −/− cells (*Figure 4B*). Several important observations arise from this analysis. First, the basal activity of the *CDKN1A* locus is higher than most pro-apoptotic genes even in p53 −/− cells, which agrees with previous studies revealing the action of strong core promoter elements at this locus (*Espinosa and Emerson, 2001*; *Morachis et al., 2010*). However, the pro-apoptotic genes *TRAF4* and *AEN* are exceptions to this trend, as they display higher basal activity in p53 −/− than *CDKN1A* and most pro-survival genes. Second, p53 action reinforces the distinction between the two classes by leading to 'super-activation' (i.e., greater than fivefold) of the survival genes *CDKN1A*, *GDF15*, *BTG2* and *MDM2* (*Figure 4C*). Although select apoptotic genes such as *TP53I3*, *PHLDA3* and *FAS* also undergo super-activation, this does not suffice to override their overall lower transcriptional output. In sum, as a group, survival genes tend to be transcribed at a higher extent than apoptotic genes, which is due to a combination of p53-dependent and -independent mechanisms.

Next, we investigated whether RNAPII pausing exerted gene-specific effects within the p53 transcriptional program. Using ChIP assays, we previously reported that the *CDKN1A* promoter carries significantly higher levels of promoter-bound RNAPII than the apoptotic genes *TP53I3*, *TNFRSF10B*, *BBC3* and *FAS* prior to p53 activation (*Espinosa et al., 2003*). GRO-seq confirms that the promoters of survival genes, including *CDKN1A*, indeed carry more transcriptionally engaged RNAPII than the promoters of apoptotic genes (*Figure 4D*); however, there was no obvious correlation between the amount of active RNAPII over the promoter and the degree of transcriptional output or induction. RNAPII pausing was proposed to modulate the timing of signal-induced gene expression, such as during the cellular response to LPS, where primary response genes were found to carry more paused RNAPII than secondary response genes (*Hargreaves et al., 2009*). However, others found that RNAPII pausing is not necessary for rapid gene induction (*Hah et al., 2011*). To investigate this issue more thoroughly within the p53 network, we performed an analysis of pausing indices as previously described (*Core et al., 2008*). This exercise revealed that although p53 target genes vary widely in their pausing indexes, there is no obvious correlation between pausing and the degree of transcriptional induction at the 1 hr time point of Nutlin treatment (*Figure 4—figure supplement 1A*). For example, the *POLH* gene, which shows a very high pausing index, displays a lower fold induction and overall lower transcriptional output than the *PHLDA3* gene, which shows little signs of RNAPII pausing (*Figure 4—figure supplement 1A*). Thus, RNAPII pausing is not a pre-requisite for rapid induction within the p53 transcriptional program. The first GRO-seq analysis in human cells revealed widespread divergent transcription at most active promoters (*Core et al., 2008*). We observed that the degree of divergence is highly variable across promoters of p53 target genes. For example, the *DRAM1* promoter displays more divergent transcription than sense transcription (*Figure 4—figure supplement 1B*), yet divergent transcription is minor at gene loci such as *CDKN1A* or *DDB2* (*Figure 1A*, *Figure 4—figure supplement 1B*, respectively). Using a 'divergence index', which evaluates the ratio of productive (sense) vs divergent transcription, we ranked p53 target genes from high to low promoter divergence, which revealed that there is no correlation between divergence and fold induction (*Figure 4—figure supplement 1B*). Thus, divergence of RNAPII in the unproductive direction does not obviously affect p53 transactivation of its target genes.

Overall, we did not find evidence of an overarching regulatory feature discriminating between survival vs apoptotic genes. Instead, GRO-seq analysis revealed many instances of potential for gene-specific regulation at key loci. For example, the *APAF1* gene, encoding a core component of the apoptosome, is activated from a non-overlapping bidirectional promoter with the *IKBIP* gene, where the two start sites are less than 280 bp apart (*Figure 4—figure supplement 2A*). The *FAS* gene, which

encodes a death receptor involved in p53-dependent apoptosis, is transcribed from a bidirectional overlapping promoter with the *ACTA2* gene, where the start site of one gene resides within the first exon of the other gene, with the two transcription units overlapping for ~850 bp (*Figure 4—figure supplement 2B*). Several p53 target genes are organized in clusters, where a group of genes within a chromatin domain are concurrently upregulated. One example is the *TNFRS10* cluster encoding the pro-apoptotic TRAIL receptors DR4 (*TNFRSF10A)* and DR5/Killer (*TNFRSF10B*), as well as the counter-acting decoy receptors DCR1 (*TNFRSF10C)* and DCR2 (*TNFRSF10D*) (*Figure 4—figure supplement 2C*). Another interesting feature revealed by GRO-seq is the presence of significant intragenic antisense transcription at many p53 target genes. In many cases such antisense transcription can be attributed to eRNAs produced from intronic p53 enhancers (see later, *Figure 5*); however, antisense transcription often originates from an overlapping transcriptional unit that is neither derived from a p53 binding event nor induced by p53. One example of this scenario is the pro-survival p53 target gene *GPR87*, which is embedded within an intron of *MED12L*, which is not a p53 target (*Figure 4—figure supplement 2D*). Thus, our GRO-seq dataset paves the way for many future mechanistic studies aimed at deciphering the impact of these gene-specific events in the regulation of p53 target genes and the cellular outcome upon p53 activation.

## Direct p53 target genes harbor highly transcribed p53 binding sites

Recently, a novel class of non-coding RNAs has been identified whose transcription originates from active enhancers. These enhancer-derived RNAs (eRNAs) have been shown to contribute to gene activation in a variety of systems (*Jiao and Slack, 2013*). A recent report characterized eRNAs derived from three distal p53 enhancers and showed that they are required for efficient p53 transactivation of neighboring genes (*Melo et al., 2013*). In order to investigate the prevalence of transcriptionally active enhancers within the p53 transcriptional program, we examined our GRO-seq data with respect to hundreds of p53 binding events as defined by ChIP-seq. Of note, we have not employed here data on histone marks or p300 occupancy to define how many of these p53 binding events reside within regions harboring the accepted hallmarks of enhancers, and thus some of these p53 binding sites should be considered as putative enhancers.

GRO-seq readily detects RNAs originating from most p53 binding events, which we refer hereto as eRNAs. A typical example is shown for the *DDIT4* locus in *Figure 5A*, where a distal p53 binding site located downstream of the gene is clearly transcribed in both the sense and antisense directions, with increased signals upon p53 activation. Interestingly, this p53RE is also transcribed in p53 −/− cells (*Figure 5A*, top track, arrow). Analysis of the *CDKN1A* locus shows transcription from the well characterized p53REs at −1.3 and −2.4 kb (*Figure 5—figure supplement 1A*). Analysis of the distal upstream region in this locus encoding the long intragenic ncRNA known as lincRNA-p21 shows transcription in both strands originating from a p53 binding site, with the antisense strand corresponding to the reported lncRNA-p21 sequence (*Figure 5—figure supplement 1B*). This suggests that lncRNA-p21 could be classified as an eRNA, as it originates from the vicinity of a p53RE associated to a canonical p53 target gene. Once again, transcripts derived from the lincRNA-p21 region can also be detected in p53 −/− cells (*Figure 5—figure supplement 1B*, top track). A rare example of a p53RE near a target gene not transcribed in p53 −/− cells is that of the *DRAM1* locus, which displays transcription of bidirectional eRNAs in p53 +/+ cells before p53 activation, with signals increasing upon Nutlin treatment (*Figure 5—figure supplement 1C*).

Analysis of the spatial distribution of p53 binding events relative to transcription start sites (TSSs) shows that direct p53 target genes display an enrichment in p53 binding close to promoters, but also within genes (*Figure 5B*). In fact, it has been estimated that ~40% of p53 enhancers are intragenic (*Nikulenkov et al., 2012*; *Menendez et al., 2013*; *Schlereth et al., 2013*; *Wang et al., 2013*). Although eRNAs derived from the sense strands can not be distinguished from the protein coding pre-mRNAs at these locations, the eRNAs arising from the antisense strands are clearly distinguishable, as illustrated for the *SYTL* and *BTG2* loci (*Figure 5C*, *Figure 5—figure supplement 1D*, respectively). Thus, p53 activation leads to antisense transcription within a large fraction of its direct target genes concurrently with activation of the protein-coding RNAs, a phenomenon with potential regulatory consequences.

Next, we analyzed the production of eRNAs at three different sets of p53 binding events: (a) distal binding sites (>25 kb of any gene), (b) proximal binding sites associated with a gene not activated by p53 (<25 kb of non GRO-seq target gene), and (c) proximal binding sites associated with a p53 target

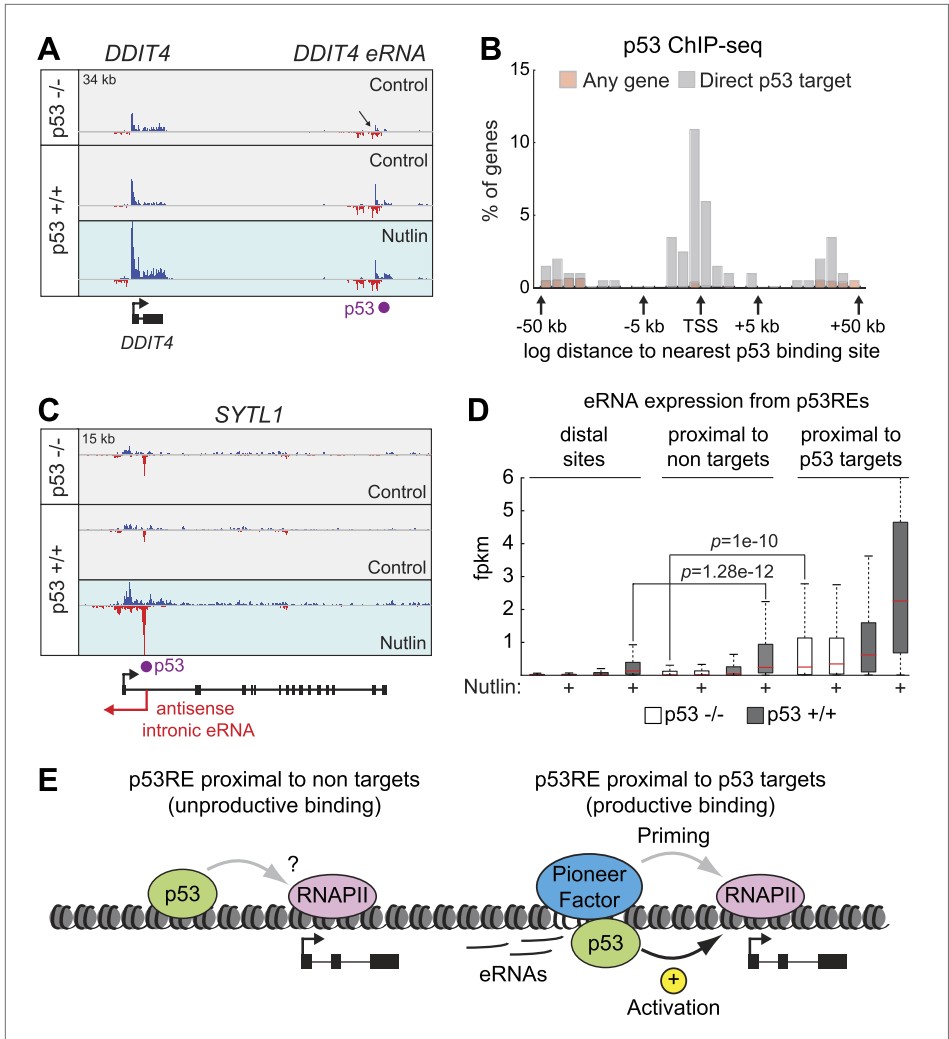

**Figure 5**. Direct p53 target genes harbor pre-activated enhancers. (**A**) GRO-seq results for the *DDIT4* locus, representative of p53 target genes that display bidirectional eRNA transcription (arrow) arising near sites of p53 binding (indicated by a purple dot). (**B**) Analysis of nearest p53 binding events relative to the transcription start site (TSS) of direct p53 target genes detected by GRO-seq (grey bars) vs all RefSeq genes (pink). (**C**) GRO-seq results for the *SYTL1* locus, representative of p53 target genes that display intronic antisense eRNA transcription arising near sites of p53 binding. (**D**) Analysis of eRNA transcription at distal p53 binding sites (>25 kb of any gene), proximal sites associate with a gene not activated by p53 (<25 kb of non-target), and those proximal to a p53 target gene identified by GRO-seq. (**E**) p53 binding sites near target genes have higher transcription levels than sites near other genes even in p53 null cells, indicating the likely action of pioneer factors.

The following figure supplements are available for figure 5:

**Figure supplement 1**. p53 stimulates eRNA production at extragenic and intragenic locations.

gene (<25 kb of a p53 target defined by GRO-seq). First, we asked what fraction of sites in each category is transcriptionally active in the four experimental conditions. This revealed that ~40% of distal sites and those not associated with p53 targets are transcriptionally active in p53 −/− cells (*Figure 5—figure supplement 1E*). The number increases to ~60% in p53 +/+ cells, likely revealing the action of basal p53 levels, as observed for the *DRAM1* locus. Expectedly, Nutlin leads to further increase in the fraction of active enhancers only in p53 +/+ cells. When the analysis is restricted to those p53REs within 25 kb of a direct p53 target, the fraction that are transcribed in p53 −/− cells increases to ~70%, and nearly all of them are transcriptionally active in Nutlin-treated p53 +/+ cells (*Figure 5—figure supplement 1E*). Thus, eRNAs are a hallmark of active p53 binding sites. We next investigated whether

there are differences across p53 binding events found in the various locations in terms of eRNA activation and overall expression. A fold induction analysis shows that p53 stimulates eRNA transcription to a similar degree regardless of location (*Figure 5—figure supplement 1F*). This result counters the notion that p53 acts as transcriptional repressor from distal sites by a mechanism described as 'enhancer interference' (*Li et al., 2012*). Interestingly, the absolute amount of eRNA produced varies greatly with location (*Figure 5D*). First, eRNA transcription is much weaker from distal sites relative to proximal sites (p=1.28e−12 for Nutlin-treated p53 +/+ cells). This could be explained by more efficient communication between enhancers and promoters when in closer proximity. Second, eRNA transcription is significantly higher from proximal sites associated with direct p53 target genes. Strikingly, this difference is already evident in p53 −/− cells (p=1e−10). Thus, a distinctive feature between direct p53 target genes and those genes proximal to bound p53REs but not activated is the strength of eRNA production in the absence of p53. In other words, p53 seems to activate gene expression at genomic locations carrying 'primed', pre-activated enhancers, likely revealing the action of pioneer factors (*Figure 5E*).

## Discussion

The importance of p53 in cancer biology is undisputed, yet the mechanisms by which this transcription factor suppresses tumor growth remain to be fully elucidated. In particular, it is unclear which p53 target genes contribute to tumor suppression in various contexts. A thorough analysis of the literature up to 2008 revealed ~120 direct p53 target genes (*Riley et al., 2008*). Since then, genomics experiments using microarrays and ChIP-seq suggest thousands of p53 targets, but very few genes were commonly identified by multiple studies (*Figure 2—figure supplement 1A,B*) (*Nikulenkov et al., 2012*; *Menendez et al., 2013*; *Schlereth et al., 2013*; *Wang et al., 2013*). The lack of overlap between these reports could be due to the fact that they employed different cell types and somewhat different experimental platforms. However, measurements of RNA steady state levels may produce a misleading view of direct p53 action, as they confound direct vs indirect effects. Thus, it is possible that cell type-specific secondary effects and post-transcriptional regulatory mechanisms strongly qualify the direct transcriptional response in different cell types. Ascertaining direct p53 action by the mere presence of a p53 binding event within an arbitrary distance to a putative target gene is an imprecise exercise, as p53 may act directly from very distal sites outside of this arbitrary cutoff (leading to false negatives) and because many proximal p53 binding events may be unproductive (leading to false positives). Because of these caveats, we investigated direct transcriptional regulation by p53 using GRO-seq.

A surprising result from our GRO-seq analysis is that a short time point of MDM2 inhibition suffices for p53 to activate hundreds of genomic loci, even prior to any detectable increase in total p53 levels. Because MDM2 functions as an E3 ligase targeting p53 for degradation (*Kubbutat et al., 1997*), there was no guarantee that the low basal levels of p53 present in a proliferating cell culture would suffice to induce transcription of its target genes. Importantly, ChIP assays demonstrate that p53 and MDM2 occupy p53REs in proliferating cells and that MDM2 binding is decreased upon Nutlin treatment (*Figure 3E*). These observations emphasize the role of MDM2 as a transcriptional repressor by masking of the p53 transactivation domains (*Oliner et al., 1993*), but do not negate the importance of p53 degradation as a repressive mechanism, as it is possible that increased p53 levels are required for activation of target genes at later time points. Our results contrast the notion that apoptotic genes require higher levels of p53 for transactivation or that they are transcriptionally induced at later time points, highlighting instead the 'primed' nature of a multifunctional p53 transcriptional response. Furthermore, this confirms that the failure of many cell types to undergo apoptosis upon Nutlin treatment is not due to a defect in transactivation of key apoptotic genes (*Henry et al., 2012*; *Sullivan et al., 2012*).

Although p53 action leads to massive gene repression at a global scale over time, it is unclear how much of these effects are direct vs indirect. Previous genomics experiments identified dozens of genes that are both bound by p53 within a certain arbitrary distance and whose steady state RNA levels decrease at late time points (*Nikulenkov et al., 2012*; *Menendez et al., 2013*; *Schlereth et al., 2013*; *Wang et al., 2013*). However, meta-analysis of these reports failed to identify a single gene commonly repressed in more than one study (*Figure 2—figure supplement 1A,B*). Recent work showed that p21 is both necessary and sufficient to downregulate many genes commonly described as direct targets of p53 repression, mostly acting via E2F4 (*Benson et al., 2013*). Other cell cycle inhibitory pathways may also converge on E2F4 repressive complexes, such as the p53-inducible miRNA miR-34a, which targets the mRNAs encoding G1-S cyclins (*Lal et al., 2011*). Our data supports the notion that most repression downstream of p53 activation is indirect. First, MDM2 inhibition by 1 hr Nutlin treatment identified

only four repressed genes, none of which showed repression at the steady state levels. In contrast, a microarray experiment at 12 hr showed hundreds of downregulated genes. Analysis of this gene set strongly supports the notion that E2F4, p21, RB and miR-34a largely mediate their repression (*Figure 2—figure supplement 1C–E*).

Interestingly, GRO-seq analysis of p53 null cells revealed that p53-MDM2 complexes might directly repress transcription at a subset of p53 targets. These genes are downregulated in the presence of MDM2-bound p53 but then activated by Nutlin. These results reveal that basal amounts of p53 found in proliferating cells create an uneven landscape among its transactivation targets, pre-activating some and repressing others. Mechanistically, p53-MDM2 complexes may directly repress transcription due to the inhibitory effects of MDM2 on components of the Pre-Initiation Complex (PIC). Early work by Tjian et al. using in vitro transcription assays demonstrated a dual mechanism of transcription inhibition by MDM2 (*Thut et al., 1997*). Their biochemical assays demonstrated that MDM2 not only masks the p53 transactivation domain, but that it also represses transcription when tethered to DNA by a GAL4 DNA binding domain. They identified an inhibitory domain in MDM2 that binds to the PIC components TBP and TFIIE, and hypothesized that MDM2 could repress transcription by targeting the basal transcription machinery. Our GRO-seq results identify specific p53 targets where this mechanism could be taking place and ChIP experiments using p53 and MDM2 antibodies confirm binding of both proteins to the p53REs at these loci. In agreement with these results, others have previously demonstrated that in proliferating cells MDM2 binds to p53REs in a p53-dependent manner, and that MDM2 recruitment to chromatin can be disrupted by Nutlin or DNA damaging agents (*White et al., 2006*). Also, excess MDM2 was shown to exert uneven repressive effects on the expression of p53 target genes, independently of effects on p53 levels or chromatin binding (*Ohkubo et al., 2006*). Altogether, these data support the arising notion that MDM2 works as a gene-specific co-regulator of p53 target genes by mechanisms other than mere p53 inhibition (*Biderman et al., 2012*).

Many research efforts in the p53 field have been devoted to the characterization of regulatory mechanisms discriminating between survival and apoptotic genes. Our GRO-seq analysis reinforced the notion that *CDKN1A*, a key mediator of arrest, differs from key apoptotic genes in several aspects. *CDKN1A* has outstanding transcriptional output among p53 target genes, which is partly due to the fact that its promoter drives substantial p53-independent transcription, but also due to potent p53-dependent transactivation. In vitro transcription assays demonstrated the *CDKN1A* core promoter initiates transcription more rapidly and effectively than the *FAS* core promoter (*Morachis et al., 2010*), and GRO-seq confirms that *FAS* has weaker transcriptional output than *CDKN1A*. However, our GRO-seq analysis failed to identify a uniform criterion discriminating between the most well studied survival and apoptotic genes. To the contrary, GRO-seq revealed that each individual p53 target gene is subject to various layers of gene-specific regulatory mechanisms, including but not restricted to differential levels of p53-independent transcription, p53 transactivation potential, RNAPII pausing, promoter divergence, extragenic vs intragenic eRNAs, overlapping promoters, clustered activation and antisense transcription.

A key observation arising from our GRO-seq analysis is that p53 target genes often have 'primed' p53REs, as denoted by significantly higher levels of eRNA production in p53 null cells. We interpret this result as the action of unknown pioneering factors acting at these putative enhancers prior to p53 signaling, which would establish enhancer-promoter communication and ready these genes for further transactivation by p53 or other stimulus-induced transcription factors. This notion is supported by a recent analysis of eRNAs at three distal p53 binding sites, which were shown to be involved in long range chromatin loops independently of p53 (*Melo et al., 2013*). This model also agrees with a recent report showing that TNF-responsive enhancers are in physical contact with their target promoters prior to TNF signaling (*Jin et al., 2013*). Thus, it is likely that the p53 transcriptional program is qualified by the action of lineage-specific factors that prepare a subset of p53 enhancers in a cell type-specific manner.

Altogether, the results presented here provide a significant advance in our understanding of the p53 transcriptional program and pave the way for functional studies of novel p53 target genes and elucidation of unique regulatory mechanisms within this tumor suppressive gene network.

## Materials and methods

### Global run-on deep-sequencing

Global run-on and library preparation for sequencing were basically done as described in *Hah et al. (2011)*. GRO-seq and microarray datasets are available at Gene Expression Omnibus, data series GSE53966.

## Cell culture

HCT116 cells were grown in McCoys 5A media and passaged 2 days in a row prior to treatment. We found passaging HCT116 cells twice before the experiment resulted in less clumping of the cells and therefore better nuclei isolation. Cells were plated at a concentration of $10 \times 10^6$ on 15 cm plates and treated 24 hr later with media containing either Nutlin-3a (10 µM) or the equivalent amount of vehicle (DMSO) for 30 min or 1 hr.

## Nuclei preparation

Cells were washed 3x with ice cold PBS and then treated with 10 ml per 15 cm plate of ice-cold Lysis Buffer (10 mM Tris–HCl pH 7.4, 2 mM $MgCl_2$, 3 mM $CaCl_2$, 0.5% NP-40, 10% glycerol, 1 mM DTT, 1x Protease Inhibitor Cocktail Tablets (Roche 11,836 153 001 Germany), 4U/ml SUPERase-In) and scrapped from the plates. Cells were centrifuged 1000×$g$ for 7 min at 4°C. Supernatant was removed and pellet was resuspended in 1.5 ml of Lysis Buffer to a homogenous mixture by pipetting 20-30X before adding another 8.5 ml of Lysis Buffer. Suspension was centrifuged with a fixed angle rotor at 1000×$g$ for 7 min at 4°C. Supernatant was removed and pellet was resuspended in 1 ml of Lysis Buffer and transferred to a 1.7 ml Eppendorf tube. Suspensions were then pelleted in a microcentrifuge at 1000×$g$ for 3 min at 4°C. Next, supernatant was removed and pellets were resuspended in 500 µl of Freezing Buffer (50 mM Tris pH 8.3, 40% glycerol, 5 mM $MgCl_2$, 0.1 mM EDTA, 4U/ml SUPERase-In). Nuclei were centrifuged 2000×$g$ for 2 min at 4°C. Pellets were resuspended in 100 µl Freezing Buffer. To determine concentration, nuclei were counted from 1 µl of suspension and Freezing Buffer was added to create as many 100 µl aliquots of $5 \times 10^6$ nuclei as possible. Aliquots were quick frozen in liquid nitrogen and stored at −80°C.

## Nuclear run-on and RNA preparation

After thawing, each 100 µl aliquot of nuclei was added to 100 µl of Reaction Buffer (10 mM Tris pH 8.0, 5 mM MgCl2, 1 mM DTT, 300 mM KCl, 20 units of SUPERase-In, 1% Sarkosyl, 500 µM ATP, GTP, CTP and Br-UTP) and incubated for 5 min at 30°C. To isolate RNA, 1 ml of Trizol was added to the reaction and vortexed to homogeneity. Samples were split in half and another 500 µl of Trizol added to each half. To isolate RNA, 220 µl chloroform was added to each half sample and samples were centrifuged at max speed for 15 min. Aqueous phase was moved into a new tube and 22.5 µl of 5M NaCl was added. Samples were Acid Phenol-Chloroform extracted twice, then Chloroform extracted once. RNA was then precipitated by adding 1 µl glyco-blue and 3 volumes ice cold ethanol to each sample before storing at −20°C for 20 min or more.

## Note on phenol and chloroform extractions

The current volume of the sample is measured and then an equal volume of Phenol-Chloroform, Chloroform or Acid Phenol-Chloroform is added. Then the mixture is vortexed and centrifuged at 12000×$g$ for 15 min (Phenol-Chloroform, Acid Phenol-Chloroform) or 10 min (Chloroform) and the top aqueous layer is kept, the lower organic layer and interphase discarded. Acid Phenol-Chloroform was stored at 4°C but was brought to room temperature before use (~30 min).

## DNAse treatment and removal of 5′ phosphate groups

Samples were centrifuged at 12,000×$g$ for 10 min washed with 70% ethanol, and then centrifuged at 12,000×$g$ for 5 min again. Pellets were air dried for 2 min and resuspended in 20 µl DEPC-treated water. Samples were base-hydrolyzed with 5 µl 1M NaOH on ice for 30 min (creating an average fragment size of 150 nt). Samples were neutralized with 25 µl 1M Tris-Cl pH6.8 and then run through a BioRad P-30 column per manufacturer's protocol. Samples were DNAse-treated in 1x RQ1 DNase buffer and 3 µl DNase I (1unit/µl, # M6101; Promega, Madison, WI) at 37°C for 10 min and then run through a BioRad P-30 column per manufacturer's protocol. To each RNA sample 8.5 µl 10 × antarctic phosphatase buffer, 1 µl of SUPERase-In and 5 µl of antarctic phosphatase was added for 1 hr at 37°C, and then run through a BioRad P-30 column per manufacturer's protocol. Final volume of RNA solution was brought up to 100 µl with DEPC-treated water and 1 µl 500 mM EDTA was added.

## Anti-BrU bead preparation

To prepare beads, 60 µl Anti-BrU agarose beads (Santa Cruz Biotech, Santa Cruz, CA) were washed twice for 5 min in 500 µl of Binding Buffer (0.5 × SSPE, 1 mM EDTA, 0.05% Tween-20). After each wash buffer was removed after centrifugation at 1000×$g$ for 2 min. Beads were then blocked in 500 µl

Blocking Buffer (0.5 × SSPE, 1 mM EDTA, 0.05% Tween-20, 0.1% PVP, and 1 mg/ml Ultrapure BSA) for 1 hr. Beads were then washed twice for 5 min each in Binding Buffer. Beads were finally resuspended in 400 µl Binding Buffer.

## Nascent RNA isolation

All washes and incubations in this section were done with rotation of the tubes. RNA (100 µl) was heated to 65°C for 5 min and kept on ice and added to prepared Anti-BrU beads in 400 µl Binding Buffer for 1 hr at room temperature. BrU-labeled nascent RNA will therefore be attached to the beads at this step. Beads were then washed with multiple wash solutions for 3 min each at room temperature then centrifuged for 2 min at 12,000×g and resuspended in the next wash. Beads were washed in 1X Binding Buffer, 1X Low Salt buffer (0.2 × SSPE, 1 mM EDTA, 0.05% Tween-20), 1X High Salt Buffer (0.5% SSPE, 1 mM EDTA, 0.05% Tween-20, 150 mM NaCl) and 2X TET buffer (TE pH 7.4, 0.05% Tween-20). BrU-labeled nascent RNA was eluted at 42°C with 4 × 125 µl of Elution Buffer (5 mM Tris pH 7.5, 300 mM NaCl, 20 mM DTT, 1 mM EDTA and 0.1% SDS). RNA was then Phenol/Chloroform extracted, Chloroform extracted and precipitated with 1.0 µl glyco-blue, 15 µl of 5M NaCl, 3 volumes 100% ethanol at −20°C for more than 20 min.

## PNK treatment and second bead-binding

Samples were centrifuged for 20 min at 12,000×g then washed with 70% ethanol and then pellets were resuspended in 50 µl PNK Reaction Buffer (45 µl of DEPC water, 5.2 µl of T4 PNK buffer, 1 µl of SUPERase_In and 1 µl of T4 PNK [New England BiolabsIpswich, MA]) and incubated at 37 C for 1 hr. To this solution 225 µl water, 5 µl 500 mM EDTA and 18 µl 5M NaCl RNA were added and then the sample was Phenol/Chloroform extracted with 300 µl twice, Chloroform extracted once and precipitated with 3 volumes 100% ethanol at 20°C for more than 20 min. Entire bead binding step was then repeated again to precipitation.

## Reverse transcription

Reverse transcription was performed as follows: RNA was resuspended in 8.0 µl water and the following was added: 1 µl dNTP mix (10 mM), 2.5 µl oNTI223HIseq primer (12.5 µM) (Sequence: 5'-pGATCGTCGGA CTGTAGAACTCT/idSp/CCTTGGCACCCGAGAATTCCATTTTTTTTTTTTTTTTTTTTVN; where p indicates 5' phosphorylation,/idSp/indicates the 1',2'-Dideoxyribose modification used to introduce a stable abasic site and VN indicates degenerate nucleotides). This mix was then heated for 3 min at 75°C and chilled briefly on ice. Then 0.5 µl SuperRnaseIn, 3.75 µl 0.1M DTT, 2.5 µl 25 mM MgCl2, 5 µl 5X Reverse Transcription Buffer, and 2 µl Superscript III Reverse Transcriptase were added and the reaction was incubated at 48°C for 30 min. To eliminate excess oNTI223HIseq primer, 4 µl Exonuclease I and 3.2 µl 10X Exonuclease I Buffer were added and the reaction was incubated at 37°C for 1 hr . Finally, RNA was eliminated by adding 1.8 µl 1N NaOH and incubated for 20 min at 98°C. The reaction was then neutralized with 2 µl of 1N HCl. Next, the cDNA was Phenol:Chloroform extracted twice, chloroform extracted once and then precipitated with 300 mM NaCl and 3 volumes of ethanol.

## Size selection

cDNA was resuspended in 8 µl of water and added to 20 µl FLB (80% Formamide, 10 mM EDTA, 1 mg/ml Xylene Cyanol, 1 mg/ml Bromophenol Blue) before loading on an 8% Urea gel. RNAs between 200–650 nt were selected and gel fragments were shattered, eluted from the gel via rotating overnight in 150 mM NaCl, 1x TE and 0.1% Tween. Entire solution was than ran through Spin X column (CLS8163; Sigma-Corning, Pittston, PA) at 10,000 RPM for 2 min at room temperature. Samples were snap frozen in liquid nitrogen and volume was reduced by spinning in speed-vac until under 500 µl. The cDNA was then Phenol:Chloroform extracted twice, Chloroform extracted once and then precipitated with 300 mM NaCl and 3 volumes of ethanol.

## Circularization and relinearization

cDNA was resuspended in 8 µl and added to 1 µl Circuligase Buffer, 0.5 µl mM ATP and 0.5 µl of 50 mM MnCl2 and 0.5 µl CicLigase (CL4111K; Epicentre, Madison, WI) and incubated at 60°C for 1 hr. Then reaction was heat-inactivated at 80°C for 20 min. To the above reaction 3.8 µl of 4X relinearization supplement (100 mM KCl, 2 mM DTT) and 1.5 µl ApeI was added and the reaction was incubated at 37°C for 1 hr. The cDNA was then Phenol:Chloroform extracted twice, Chloroform extracted once and then precipitated with 300 mM NaCl and 3 volumes of ethanol.

## PCR and sequencing

cDNA was resuspended in 15 µl of water and added to 10 µl 5x Phusion HF buffer, 0.5 µl 25 µM HighSeq fwd primer, 0.5 µl 25 µM HighSeq rev primer, 2.5 µl 10 mM dNTPs, 1 µl Phusion DNA polymerase and 20 µl water. PCR amplification was done as follows: Step 1: 98°C for 3 min, Step 2: 98°C for 80 s Step 3: 60°C for 90 s, Step 4: 72°C for 30 s, Step 5: repeat Steps 2–4 for 15–20 cycles, dependent on concentration of the final cDNA. Step 6. 65°C for 10 min, Step 7: 4°C. The entire PCR reaction was then Phenol-Chloroform extracted twice, Chloroform extracted once and precipitated with 300 mM NaCl and 3 volumes of ethanol. The DNA was then run on a 6% native gel (1x TBE, 6% acrylamide [19:1]). Products between 250–650 nt were gel extracted and precipitated as described above. Samples were then sequenced using an Illumina HiSeq2000 instrument with 50 bp reads.

## Computational analysis

Unless otherwise noted all processing was done with python version 2.6. Graphs were created with either Microsoft Excel or the python package matplotlib version 1.0.1.

### Mapping

Two biological replicates of Nutlin-treated and DMSO-treated (control) samples were combined. Single samples were created for p53 null cells and for the 30 min Nutlin treatment. Total reads were as follows p53 +/+ control (DMSO 1 hr) = 214019557 (69399243 + 144620314), p53 +/+ Nutlin 1 hr= 325026803 (166083763 + 158943040), p53 −/− control (DMSO 1 hr) = 184022999, p53 −/− Nutlin 1 hr = 162410974, p53 +/+ Nutlin 30 min = 205211352, p53 +/+ Control (DMSO 30 min) = 175418701. With regards to the quality of the bases in each sample, the median Illumina quality score for all bases was always greater than 32, most of them above 39, that is the inferred base call accuracy was greater than 99.9% for all bases in all samples (Fastx v. 0.0.13.2). Reads were mapped to the human genome (hg19 downloaded from bowtie website) using bowtie (version 0.12.7) with the command bowtie -S -t -v 2 –-best. Using this strategy approximately 70% of the reads mapped. Mapping created a sam file that was then processed by samtools view -bS–o and samtools sort to create a sorted bam file (version 0.1.16).

### Creation of igv figures

The sorted bam files mentioned above were used to make a Bedgraph by running genomeCoverageBed–bg (v2.12.0) on each strand separately. The negative strand values were changed to negative values in the BedGraph. The BedGraph values were then divided by the number of million of mappable reads. The two files (strands) were concatenated back together and igvtools sort and igvtools tile was used to create a tdf file that was loaded into igv for creation of snapshots of genes (IGVtools 1.5.10, IGV version 2.0.34).

### Calculation of activities and pausing indexes

Calculations were done exactly as in *Core et al. (2008)* unless otherwise noted. Gene annotations (hg19) were downloaded from: http://hgdownload.cse.ucsc.edu/goldenPath/hg19/database/refGene.txt.gz.

Number of reads in the gene body (1 kb from transcription start site [TSS] to the end of the annotation) and number of reads around the promoter (−100 to +400 bp from annotated TSS) were counted by the program coverageBed v2.12.0. A program to calculate fpkm, pausing indexes, gene activity, and promoter activity was written and run on python 2.6. Fisher's exact test was done using the python module fisher 0.1.4 downloaded from https://pypi.python.org/pypi/fisher/. RefSeq genes shorter than 1 kb were not used. Genes that are differentially expressed were determined in R version 2.13.0 using DEseq v1.4.1 (*Anders and Huber, 2010*). Settings for DEseq were cds <–estimateSizeFactors(cds), method = 'blind', sharingMode = 'fit-only'. Genes were called as differentially transcribed if they had an adjusted p-value less than or equal to 0.1. Manual curation was used to choose the most parsimonious isoform for the Nutlin vs control (DMSO) comparisons. For genes only differentially expressed across cell lines, we utilized the isoform with the highest fold change (p53+/+ control vs p53 −/− controls). For all other genes we used the isoform identifier with the highest fold change between p53+/+ control and p53+/+ Nutlin.

### Microarray analysis

HCT116 cells were grown in McCoy's 5A and passaged the day prior to treatment. Cells were plated at a concentration of 300,000 cells per well of six well plate and treated 24 hr later with either Nutlin-3

(10 µM) or the equivalent amount of vehicle (DMSO) for 12 hr. Total RNA from HCT116 cells was harvested with an RNeasy kit (Qiagen, Germantown, MD) and analyzed on Affymetrix HuGene 1.0 ST arrays following the manufacturer's instructions. Microarray data were processed using Partek Genomics Suite 6.6. Anova was used to call differentially expressed genes for which any isoform showed a fold change > |+/−1.5| with FDR <0.05. There were 362 genes called as upregulated and 367 genes as downregulated.

## Comparative analysis of GRO-seq vs microarray data

The microarray analysis provided a list of gene names and their fold change on the microarray. Since many of the genes had multiple isoforms we simplified by keeping only the isoform with the greatest fold change between Control and Nutlin. For comparisons of microarray and GRO-seq, a list of genes common to both analyses was used. If a gene was found in only one analysis (GRO-seq or microarray) it was not used. In the microarray graphs, expression values from the three biological replicates were averaged. Graphs (MAplot, scatter plot, box and wiskers) were created in python by using matplotlib.

## Meta-analysis of published p53 ChIP-seq data

To create a list of high confidence p53 binding sites, we combined the data from of 7 ChIP assays for p53 (*Wei et al., 2006*; *Smeenk et al., 2008*; *Smeenk et al., 2011*; *Nikulenkov et al., 2012*) and kept only sites that were found in at least five of the seven assays. The assays covered three cell lines (HCT116, U20S, MCF7) and 6 different conditions (untreated, Nutlin, 5FU, RITA, ActD, Etoposide). First, all samples were lifted over to hg19 coordinates. Second, a file was created that contained all sites from each experiment and the sites were then sorted and merged using BedTools (2.16.1). This created a list of 106,600 total sites, only 1481 of which were found in 5/7 datasets.

## Meta-analysis of published investigations of the p53 transcriptional program using a combination of microarray and ChIP-seq data

To generate the Venn diagrams in *Figure 2—figure supplement 1B*, we analyzed the information provided in the following papers: (*Nikulenkov et al., 2012*; *Menendez et al., 2013*; *Schlereth et al., 2013*; *Wang et al., 2013*). All gene lists are provided in *Supplementary file 2*. Genes bound by p53 and upregulated (339) or downregulated (8) upon Nutlin treatment in MCF7 cells were obtained from Tables S7 and S10 of the Nikulenkov et al paper. Genes bound by p53 and upregulated or downregulated by either Nutlin or Doxorubicin treatment in U2OS cells were obtained from Supplemental Dataset 3, (file: nar-00620-v-2013-File011), of the Menendez et al paper. We removed from the analysis of 'p53 activated genes' those that were repressed by Nutlin or Doxorubicin, keeping only those that were either activated by both drugs or activated by one drug and unchanged with the other. Genes bound by p53 and upregulated or downregulated upon p53 overexpression in SAOS2 cells were obtained from Table S1 of the Schlereth et al paper. We removed from this list duplicate gene entries. Genes bound by p53 and upregulated or downregulated upon p53 overexpression in HCT116 p53 −/− cells were obtained from Table S3 and Table S5 of the Wang et al paper. We manually curated all gene lists to ensure that gene nomenclature was the same before analyzing the overlap among the various lists using Venny: http://bioinfogp.cnb.csic.es/tools/venny/index.html. Note: because some these published studies used somewhat different microarray platforms, it is possible that some genes were not actually measured in all four studies.

## Analysis of eRNAs derived from p53 binding sites

For each merged ChIP site mentioned above a bed file was created that contain two copies of each site, one assigned to each strand (213,200 'stranded-sites'). Then sites were removed if overlapped with an annotated refseq gene on the same strand (160,604 stranded-sites left after removal). The number of reads covering each remaining site were counted using coveragebed and fpkm calculate for each site. To create a list of high confidence p53 binding sites we could use for analyses of eRNAs we used only sites that were bound in five of the seven ChIP experiments used above (2283 stranded-sites, 1454 sites). We then divided these sites into three categories. Distal sites (more than 25 kb from any annotated refseq gene) (992 stranded-sites, 496 sites); proximal to non-target gene sites (less than or equal to 25 kb from a gene i.e. not a GRO-seq target, 1187 stranded-sites, 555 sites with only one strand, 316 sites with both strands); and sites near p53 target genes (less than or equal to 25 kb from a gene i.e. a GRO-seq target, 104 stranded-sites, 70 sites with only one strand, 17 sites with both strands). We then determined a percent of the sites that are active by the same method used in *Core*

*et al. (2008)*. Briefly, background was set at 3% of the reads. We estimate the probability of observing at least N reads in an interval of length L using a Poisson distribution. If the probability was above 0.01 the site was called transcriptionally active. For each site, if either of the two strands was transcriptionally active (logical or) the site was counted as active therefore for the *Figure 5—figure supplement 1E*, n = 496, 496, 496, 496, 871, 871, 871, 871, 87, 87, 87, 87. For *Figure 5D* we wanted to include information from both strands when available so stranded-sites were used to determine the fpkm of each site, therefore n = 992, 992, 992, 992, 1187, 1187, 1187, 1187, 104, 104, 104, 104. Similarly, for *Figure 4—figure supplement 1F* we included all stranded-sites but for each comparison we had to remove any sites in a given sample that had 0 reads, therefore n = 323, 602, 566, 897, 83, 100.

## Note on distance to the nearest binding site

To determine the distance to the nearest p53 binding site, for all genes the program pybedtools and the script closest was used. The sites were the 1481 sites that were in five of seven ChIP experiments as mentioned above. The target genes were the 202 up and down-regulated genes by GRO-seq. The distances were then binned to create the histogram shown in *Figure 5B*. The 10 most outer bins on the left and right are bins of 5 kb; the inner bins are bins of 1 kb.

## Overlap of genes downregulated in microarray and miR-34a targets

The published genes that were downregualted upon miR-34a overexpression in HCT116 cells (*Lal et al., 2011*) (2091 total, 1765 also found in our microarray experiment) were compared to the genes that were downregulated upon Nutlin treatment in our microarray experiment (367 total, 342 also found in the published microarray by *Lal et al. (2011)*). Of the 342, 245 (72%) were downregulated in the miR-34a overexpression experiment. All genes which overlap (16,553) between the two microarrays (miR-34a overexpression n = 21,050 and Nutlin treatment, n = 19,901) were determined assuming Lal et al used the annotations from version 32 of Affymetrix U133 plus 2.0 mRNA microarray. Hypergeometric was used to calculate a p-value.

## Ingenuity pathway analysis (IPA) of genes downregulated upon Nutlin treatment

The 367 genes shown to be downregulated upon Nutlin treatment in our microarray experiment ('Microarray analysis') were subjected to IPA Upstream Regulator Analysis, which identifies upstream transcriptional regulators that can explain the observed gene expression changes in a user's dataset. The top three upstream transcriptional regulators identified in our dataset were E2F4, CDKN1A and RB, all three identified as 'transcriptional repressors' by this analysis. Statistical significance and p-values were determined by IPA using a Fisher's Exact Test. Detailed explanation of this analysis is provided by IPA at: http://pages.ingenuity.com/IngenuityUpstreamRegulatorAnalysisWhitepaper.html.

## Oncomine analysis of p53 wild type and p53 mutant cell lines and tumors

Oncomine (Compendia Bioscience, Ann Arbor, MI) was used for analysis and visualization of expression data from the Garnett Cell Line dataset (*Garnett et al., 2012*) containing gene expression data for hundreds of p53 wild type and p53 mutant cancer cell lines or the Ivshina Breast Carcinoma dataset (*Ivshina et al., 2006*). Filters in the Oncomine database were set to select Garnett Cell Line dataset (or the Ivshina Breast Carcinoma dataset), and *TP53* mutation status. Genes analyzed were individually filtered with the above filters in place, and analysis was adjusted to view under-expression with p53mt vs p53wt (for basal up with p53 genes) and adjusted to view over-expression with p53mt vs p53wt (for basal down with p53 genes). Graphs represent the log2 median centered intensity (MCI) of gene expression from selected genes on the array. The p-value is derived from a Student's *t* test between p53wt and p53mt expression, and the gene rank reveals the rank of that gene in the dataset among all other genes in the analysis according to the p-value for the differential expression analysis.

## Gene set enrichment analysis (GSEA) of GRO-seq p53 target genes analyzed by Oncomine

To generate the GSEA plots shown in *Figure 3—figure supplement 2A*, we analyzed the position of 183 GRO-seq p53 target genes (out of 198) which were also present in the gene rank generated by Oncomine analysis of the Garnett Cell Line dataset (see above) using GSEA software from the Broad

Institute: (http://www.broadinstitute.org/gsea/index.jsp). In this analysis, the 'molecular profile data' was the Oncomine gene rank and the 'gene set database' was the list the GRO-seq p53 target genes. The analysis was repeated using the 'microarray only' genes (i.e., genes upregulated by Nutlin as seen only in our microarray experiment). For the scatter plot log2 fold changes in the Oncomine dataset were plotted against fold changes in GRO-seq data using R.

## RT-PCR

Total RNA was isolated using Trizol (Life Technologies, Frederick, MD) following the manufacturer's instructions. cDNA was generated using qScript kit (Quanta Biosciences, Gaithersburg, MD) with random priming. cDNA was subjected to standard or quantitative PCR (Q-PCR) using SYBR green or SYBR select master mix on a Viia 7 instrument (Life Technologies) with the primer pairs listed below. The 18S rRNA was used for normalization. Experiments were done in biological triplicate and error bars represent the SEM.

## Primers used for Q-RT-PCR

Gene, Forward Primer, Reverse Primer
Gene, Forward Primer, Reverse Primer
**18S rRNA,** GCCGCTAGAGGTGAAATTCTTG, CTTTCGCTCTGGTCCGTCTT
**ADAMTS7**, GTCGAGCCTCCCCGCTGTG, CAGCGTCTCGCAGAACCCGA
**ALOX5**, AAACGAGCTGTTCCTGGGCATGT, GGCCTCGAGGTTCTTGCGGA
**APOBEC3C**, AGTATCCATGTTACCAGGAGGGGCT, TTTAAAATCTTCATAGTCCATGATCTCCACAGCGA
**ASCC3**, GCTTGCAGAGAGTCCACTTTGGGT, ACAGCTCTCTGGCTTTCCCTTGTG
**ASS1**, TGAGGAGCTGGTGAGCATGAACG, ACCCGGTGGCATCAGTTGGC
**ASTN2**, TTTTCTGCCGCAGCGAGGAGG, AGAGTCAAATAATATACGTGATTTTGGTGTCCTTGA
**BLOC1S2**, CCGAGGGCGTACTGGCGAC, GGCATCGTCTCGGGCGGG
**C19ORF82 (LOC284385),** ATTTAGAGGAGGGACCCGGC, AAGCTTTGGAGGAGCATCCC
**CDC42BPG**, TGTCCTGCCCCCAGGGATCG, GGGCCGTTCTGAGACCTGCATTAG
**CDKN1A**, CTGGAGACTCTCAGGGTCGAAA, GATTAGGGCTTCCTCTTGGAGAA
**CEP85L**, AGCTCCTTGGCAACAGCAGCAA, TGATCCAGTCAAAACCTGCATTGGTGT
**CHAC1**, TGAAGATCATGAGGGCTGCACTTGG, CAGGGCCTTGCTTACCTGCTCC
**DDB2**, TCTACTCGCTGCCGCACAGG, TCGGGACTGAAACAAGCTGCGT
**EFNB1,** TGGGCAAGATCCCAATGCTGTG, TGCTTGCCATCAGAGTCACCCA
**FAM210B**, TGGCACTGTTGGCGTGTCAT, CAGGCATGTCCACACCACTTGA
**FAM212B**, AATGGGCGCATGACGATGGAGA, TGCAGTGCACCCATCATGCAGT
**GDF15**, TAACCAGGCTGCGGGCCAAC, CAGCCGCACTTCTGGCGTGA
**GJB5**, CTGCTGGGAGCCAGGAGAGC, CGCGTCAGCTGCTACTGAGTGA
**GPR87**, AACCTATGCTGAACCCACGCCT, GCCGTGCAGCTCGTTATTTGGT
**GRIN2C**, ACCGTGACATGCACACCCACAT, TGAAGGCATCCAGCTTCCCCAT
**HES2**, CTGGGCCGGGAGAACTCCAA, GCAGGAAGCGCACGGTCATT
**KCNN4**, CCGCCATCAACGCGTTCCG, CCCGGAGCTTCCGGTGTTTCA
**LRP1**, ACCCACTGAGGGGGACCATGT, TCCCATCCATCGCTGCCGTC
**PHLDA3**, TAAACCACCGGGCGCACCAT, CAGAGGGAACAACGAAGCTGCC
**PTP4A1**, TAAGACAAAAGCGGCGTGGAGC, ACGCAGCCGCATTTTAGGACG
**PTPRE**, CTTTTCCCGGCTCACCTGGTTCAG, GGGCATCTTCTTGTCGCTGGTG
**RINL**, TGGCTTCCTGCAACCTGACGAT, TGCTTTGTTCACCTGTGGTGGG
**SCL30A1**, TTGGACCCCGCAGACCCAGA, TGGTCAGGTTCTCTGACAAGATTTCCATT
**SULF2**, CCCCCGGACTCGAAACATGGA, ACTTTCGACGCTGAAACTGCCTGT
**TM7SF3**, CTTTCACAAATGTGCCTTTTCAAACTAATGACTTC, AGCATGCCCCATACTGCCAGG
**TOB1**, GAAGCAGCCCCTTCCCAGGAG, GCTTTTCAGGATACCAGTGCCCTTCAT
**TP53INP1**, AATGAGAAAGAAGATGATGAATGGATT, TGCTGAGAAACCAGTGCAAGTATC
**WRAP73**, TGCCTGGGGAAGGCGACTTTG, GAGGGCCATCGAGTCTCCGC

## Western blotting

Protein extracts were separated by SDS-PAGE and transferred to PVDF membranes. Blots were probed with the primary antibodies listed below, followed by peroxidase-conjugated secondary

antibodies (Santa Cruz Biotechnologies). Detection was by enhanced chemiluminescence or Luminata Crescendo (Millipore, Tauton, MA) and images were captured using an ImageQuant LAS4000 digital camera system (GE Healthcare, Sweden). Antibodies: p21 (sc-817; Santa Cruz); p53 (DO-1, OP43; Calbiochem, Tauton, MA); MDM2 (SMP14, sc-965; Santa cruz, used in combination with True Blot anti-mouse secondary antibody (Rockland, Boyertown, PA) to avoid IgG signal in immunopre-cipitated samples); GRIN2C (LS-C157785; LifeSpan BioSciences, Seattle, WA); PTCDH4 (sc-139478; Santa Cruz (C-15)); RINL (OAAB08343; Aviva Systems Biology, San Diego, CA); Nucleolin (sc-8031; Santa Cruz); α-tubulin (T9026; Sigma, Israel).

## BrdU incorporation assay

Cell proliferation was assessed by quantification of cells actively synthesizing DNA. HCT116 p53+/+ and HCT116 p53−/− cells treated with 10 µM Nutlin-3 as indicated. 60 min prior to harvesting cells, 1 mg/ml of BrdU was added to cultivation medium. After trypsinization, cells were fixed with 70% ethanol. DNA denaturation was achieved by incubation with 2M HCl with 0.5% Triton X-100 for 10 min at 37°C. Before staining with anti-BrdU antibody (sc-51514; Santa Cruz Biotechnology), samples were neutralized with 0.1M $Na_2B_4O_7$. Following incubation with secondary antibody (A21202, Life Technologies). BrdU positivity was analyzed by flow cytometry. Data represent average numbers of BrdU positive cells from three independent experiments (each run in duplicate). T test was used for statistical analysis; an asterisk denotes significance at p<0.001.

## ChIP assays

Chromatin immunoprecipitation (ChIP) was carried out as previously described (*Gomes et al., 2006*). Briefly, for quantitative ChIP analysis of the *GDF15 and PTP4A1* locus, cells were cross-linked with 1% formaldehyde and whole-cell lysates were prepared in RIPA buffer. 1 mg of protein extract was immu-noprecipitated with specific antibodies. Real-time PCR was carried out on ChIP-enriched DNA against a standard curve of genomic DNA, with amplicons tiling across the locus. Enrichment values for each amplicon were calculated as percentage of the amplicon with maximum signal for each antibody or expressed as equivalents of gDNA in nanograms. For MDM2 ChIPs a modified RIPA pH 7.4 buffer was used (150 mM NaCl, 1% Nonidet P-40, 0.5% sodium deoxycholate, 0.025% SDS, 50 mM Tris pH 7.6, 5 mM EDTA, protease/phosphatase inhibitors).

### Antibodies used for ChIP

RNAPII S5P (H14): Covance (Dedham, MA) MMS134R; RNAPII S2P (H5): Covance MMS129R; p53: Calbiochem OP43; MDM2: Santa Cruz sc-13161 (D-7); normal mouse IgG: Santa Cruz sc-2025.

## Primers used for ChIP

Location of amplicon center relative to TSS, forward primer, reverse primer (5'>3'):

### GDF15 tiling:

GDF15 -692, AAGGTGAGCGCTGAGCCAGA, AGCAGGGTGCTCCCAAACCC
GDF15 +54, CCAGCTCAGAGCCGCAACCT, CAGCCACGAGAGCACCAGCA
GDF15 +851, GAGGCTCCAGGGGTGCAGG, GTCCCACCACAGCCCAGACC
GDF15 +1596, GGGTGCTTCCTATTTTAGGCAGGCT, ATGGCTGGCTCCTCCTCTTCCT
GDF15 +2236, CGCCTTCACCGGGCTCTGTT, CGTGTCACGTCCCACGACCT
GDF15 +2991, TCGGGGGCTGGTCTGATGGA, CCTGGACACCACAGGGAACAGT
GDF15 +4107, TTATGCCAGGCCCCGGACAC, CCGCAAAGTGGGGTGGTCCT

### PTP4A1 tiling:

PTP4A1 -1524, AGGCTAAGACCCCCAGGCCA, ACCACAAGCAGTGCTGACACGA
PTP4A1 +16, GGGCCCAGGTGCCAAGGTAA, GGCTGGAACCGCGTCTCAGT
PTP4A1 +2484, TGGAGGAGTGTCTTGGGCGG, TGGAGCCACCTAGTGGATGGGG
PTP4A1 +4564, TGCTCCACCAAGAAGCCCCC, ACACTTCTGTGTCCAGGATAACCACTC
PTP4A1 +7208, ACTTCTGAGTTGGGGAATGTTTGGAGAA, AGTACTGGAGCTCTGGTGAGGTAAGAA
PTP4A1 +9477, ACATGCATTTATCATCATTGAGATTGGTTTGCT, AGATGAGGACACACATTAAGTGAAGTGCAA

PTP4A1 +11301, TGCCATACATGTTAATATTCTACATTCTTGCTTCCT,
TCAATTCAGGTGTGAGGCACATATATACACA
PTP4A1 +12,375, AGCCACTTGGGACAAGTCAATGCT, CGTTTTGTGCTGTGTAGGAAATACCGA

p53REs at various gene loci:

CDK1NA (p21) −2283, AGCAGGCTGTGGCTCTGATT, CAAAATAGCCACCAGCCTCTTCT
PTP4A1 +1789, AAATGGCCTGGTTCGGAGCGT, ATGTATGCTGGTGCCCGGACG
HES2 +5170, TCTCTGGGGGGCTGAGTGGGAGT, CGGGCAGCTCTCTGAAGGGCA
CDC42BPG +1373, CGGCATGGGGGCCCTCTGTT, CTCTGGCCTCCCACAGGGCAT
ADAMTS7 -48815, GTAAGGCGGGCCAAGGAAGGC, TTCAAGGGGTGGGCGCTCTG
LRP1 -694, GGGCGGGCGGGTACTAAGGT, CAGGGACACCCGAGGGGACA
TOB1 -38453, GCTGGCTACGGGCTGGTCTC, CAGGTGCTGGCTTGCTGCCC
ASS1 +116325, GTGGCTAGTGGGAGGGGGCAT, TGACAAGGGCCAGTGCTGAGGA
CEP85L +77725, AGGAGGGGAGGAGGAGATTGTTGC, GCATGTGTTCCTCAGCGTAATCTGT

## p53 N-terminus detection by flow cytometry

HCT116 cells were fixed and stained as described previously (*Andrysik, 2013*). Briefly, trypsinized cells were fixed with methanol and incubated with blocking solution (1% BSA-V in PBS/0.1% Triton X-100, 0.01% sodium azide, pH 7.2). p53 level was detected using DO-1 antibody (Calbiochem OP43) and Alexa Fluor A488 secondary antibody. At least 10,000 single cells were measured by Accuri C6 flow cytometer to asses signal distribution in cellular population.

## Immunofluorescence

Following the Nutlin treatment period, HCT116 cells cultivated on glass cover slips were fixed with cold methanol and washed briefly with wash buffer (PBS with 0.1% Triton X-100). After incubation with blocking solution (1% BSA-V in wash buffer, 0.01% sodium azide, pH 7.2) cells were labeled with p53 antibody (sc-6343X; 1:1000; Santa Cruz FL393) or isotype control. Three times washed cells were incubated with secondary antibody (1:2000; Alexa Fluor A21202; Invitrogen, Frederick, MD), nuclei were counterstained with DAPI (Invitrogen, 0.2 µg/ml) and cover slips were mounted using anti-fade medium (Fluoromount, Diagnostic Biosystems, Pleasanton, CA). Images were taken using Zeiss 510 Confocal Laser Scanning Microscope.

## Acknowledgements

This work was supported primarily by NIH RO1CA117907 grant (JME), NSF grant MCB1243522 (JME), a Butcher Foundation Seed Grant (JME, RDD), the Boettcher Foundation's Webb-Waring Biomedical Research program (RDD), a NIH training grant N 2T15 LM009451 (MAA), a post-doctoral fellowship from the American Cancer Society (HM). JME is an HHMI Early Career Scientist. We thank Dr Honnold and Dr Caldwell for their inspiring work.

## Additional information

### Competing interests

JME: Reviewing editor, *eLife*. The other authors declare that no competing interests exist.

### Funding

| Funder | Grant reference number | Author |
| --- | --- | --- |
| Howard Hughes Medical Institute | Early Career Award | Joaquin M Espinosa |
| National Institutes of Health | RO1 CA117907-07 | Joaquin M Espinosa |
| Butcher Foundation | | Robin D Dowell, Joaquin M Espinosa |
| Boettcher Foundation | | Robin D Dowell |
| National Science Foundation | MCB1243522 | Joaquin M Espinosa |
| American Cancer Society | | Hestia S Mellert |
| National Institutes of Health | 2T15 LM009451 | Mary Ann Allen |

The funders had no role in study design, data collection and interpretation, or the decision to submit the work for publication.

## Author contributions

MAA, HSM, Conception and design, Acquisition of data, Analysis and interpretation of data, Drafting or revising the article; JME, Conception and design, Acquisition of data, Analysis and interpretation of data, Drafting or revising the article; ZA, VLD, Acquisition of data, Analysis and interpretation of data; AG, Acquisition of data, Analysis and interpretation of data; JAF, MDG, RDD, Conception and design, Analysis and interpretation of data, Drafting or revising the article; KDS, Conception and design, Acquisition of data, Analysis and interpretation of data; XL, WLK, Conception and design, Drafting or revising the article, Contributed unpublished essential data or reagents

# Additional files

## Supplementary files

• Supplementary file 1. Genes upregulated at the transcriptional level in HCT116 p53 +/+ cells treated with 10 μM Nutlin-3a for 1 hr as detected by GRO-seq (198 genes). DeSeq algorithm was used to detect annotated gene loci whose GRO-seq signal was statistically different between DMSO- and Nutlin-treated cells (adjusted p<0.1). Columns in this table indicate: (a) Gene name, (b) Whether the gene was previously identified as a direct p53 target gene in the literature, (c–f) Whether the gene was predicted to be a direct p53 target gene by one or more recent studies employing ChIP-seq and microarrays (*Figure 2—figure supplements 1 and 2*), (g) fpkm in p53 +/+ control, (h) fpkm in p53 +/+ Nutlin-3, (i) Fold induction, (j) Protein Function, (k) Putative downstream pathway within the p53 network, (l) References describing the gene as a direct target, putative target or establishing gene function.

• Supplementary file 2. Lists of genes bound by p53 as defined by ChIP-seq and concurrently upregulated or downregulated as defined by microarray measurements of RNA steady state levels. Related to *Figure 2—figure supplement 1A,B*. See 'Materials and methods', 'Computational Analysis–Meta-analysis of published investigations of the p53 transcriptional program using a combination of microarray and ChIP-seq data' for details.

## Major datasets

The following dataset was generated:

| Author(s) | Year | Dataset title | Dataset ID and/or URL | Database, license, and accessibility information |
| --- | --- | --- | --- | --- |
| Allen Mary Ann, Mellert Hestia, Dengler Veronica, Andrysik Zdenek, Guarnieri Anna, Freeman Justin A, Luo Xin, Kraus W Lee, Dowell Robin D and Espinosa Joaquin M | 2014 | Global analysis of p53-regulated transcription reveals its direct targets and unexpected regulatory mechanisms | http://www.ncbi.nlm.nih.gov/geo/query/acc.cgi?acc=GSE53966 | Publicly available at NCBI Gene Expression Omnibus. |

The following previously published datasets were used:

| Author(s) | Year | Dataset title | Dataset ID and/or URL | Database, license, and accessibility information |
| --- | --- | --- | --- | --- |
| Nikulenkov F, Spinnler C, Li H, Tonelli C, Shi Y, Turunen M, Kivioja T, Ignatiev I, Kel A, Taipale J, Selivanova G | 2012 | Microarray and ChIP-seq data from Insights into p53 transcriptional function via genome-wide chromatin occupancy and gene expression analysis | SRP007261; http://www.ncbi.nlm.nih.gov/sra/SRP007261 | Publicly available at the NCBI Sequence Read Archive (http://www.ncbi.nlm.nih.gov/sra). |

| | | | | |
|---|---|---|---|---|
| Garnett MJ, Edelman EJ, Heidorn SJ, Greenman CD, Dastur A, Lau KW, Greninger P, Thompson IR, Luo X, Soares J, Liu Q, Iorio F, Surdez D, Chen L, Milano RJ, Bignell GR, Tam AT, Davies H, Stevenson JA, Barthorpe S, Lutz SR, Kogera F, Lawrence K, McLaren-Douglas A, Mitropoulos X, Mironenko T, Thi H, Richardson L, Zhou W, Jewitt F, Zhang T, O'Brien P, Boisvert JL, Price S, Hur W, Yang W, Deng X, Butler A, Choi HG, Chang JW, Baselga J, Stamenkovic I, Engelman JA, Sharma SV, Delattre O, Saez-Rodriguez J, Gray NS, Settleman J, Futreal PA, Haber DA, Stratton MR, Ramaswamy S, McDermott U, Benes CH | 2011 | Gene expression analysis of 789 cancer cell lines using the Affymetrix HT-HG-U133A v2 platform | E-MTAB-783; http://www.ebi.ac.uk/arrayexpress/experiments/E-MTAB-783/ | Publicly available at ArrayExpress (http://www.ebi.ac.uk/arrayexpress). |
| Smeenk L, van Heeringen SJ, Koeppel M, van Driel MA, Bartels SJ, Akkers RC, Denissov S, Stunnenberg HG, Lohrum M | 2008 | Chromatin immunoprecipitation of p53 in human osteocarcoma cells | E-TABM-442; http://www.ebi.ac.uk/arrayexpress/experiments/E-TABM-442/ | Publicly available at ArrayExpress (http://www.ebi.ac.uk/arrayexpress). |
| Wei CL, Wu Q, Vega VB, Chiu KP, Ng P, Zhang T, Shahab A, Yong HC, Fu Y, Weng Z, Liu J, Zhao XD, Chew JL, Lee YL, Kuznetsov VA, Sung WK, Miller LD, Lim B, Liu ET, Yu Q, Ng HH, Ruan Y | 2006 | p53 ChIP data from A global map of p53 transcription-factor binding sites in the human genome | http://hgdownload.cse.ucsc.edu/goldenPath/hg17/encode/database/encodeGisChipPet.txt.gz | Available at http://hgdownload.cse.ucsc.edu/downloads.html. |

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
