## [Decision Letter]

Thank you for sending your work entitled “Global analysis of p53-regulated transcription identifies its direct targets and unexpected regulatory mechanisms” for consideration at *eLife*. Your article has been favorably evaluated by a Senior editor and 3 reviewers, one of whom is a member of our Board of Reviewing Editors.

The Reviewing editor and the other reviewers discussed their comments before we reached this decision, and the Reviewing editor has assembled the following comments to help you prepare a revised submission.

The three referees agree that this is an interesting study that could be appropriate in principle for publication in *eLife*. It was felt, however, that although the study has some very valuable components, some more indepth analysis is needed. The two most interesting findings to address with more information and validation relate to the observation that Mdm2 is constitutively repressing p53 activation of some of its targets, and that GRO-seq analysis reveals several new targets of p53. The authors are requested to respond to the following comments.

Major comments:

1) It is possible that Nutlin does increase p53 levels in a small percent of the cells by 1 hour and it is those few cells with high p53 that have the increased transcription of p53 targets. Can the authors rule this out? Have they performed immunofluorescence analysis to see whether there are some ’jackpot‘ cells where p53 protein is rapidly induced by Nutlin?

2) While the Nutlin treatment allows direct ability to assay p53 function, a second method of p53 induction would greatly enhance these findings. While not direct necessarily, does an oncogenic or DNA-damage insult induce some of these new target genes (even if a few were assayed by qPCR on a gene by gene basis)? This would also address the survival vs. apoptotic regulation, since Nutlin treatment may have bias towards survival.

3) The authors should perform an Mdm2 ChIP at a few select targets such as p21 to see whether the Mdm2 signal is down by 1 hour. This is an important experiment because it would extend and confirm the most interesting aspect of this study. Endogenously expressed Mdm2 ChIPs are not very commonly reported in the literature, but have been accomplished. See for example, papers by J. Bargonetti and colleagues (White et al., Cancer Research 2006; Arva et al., JBC 2005).

4) Relatedly, in the Results section, the authors conclude that p53 acts as a repressor at a subset of genes under basal conditions (also shown in supplemental Figure 3). This is based on the comparison of expression in the p53 wt vs. null cells. Even though these genes are considered 'direct' targets from GRO-seq analysis after nutlin treatment, it is unclear whether they have evidence of p53 binding this small subset of genes, and since this is not an immediate knockdown experiment, should the authors consider secondary effects as they did in the first part of the paper? To support a direct role for p53 in repressing these promoters, the authors could ask whether p53 and/or MDM2 are at these basally regulated promoters by performing ChIP-qPCR or by using published ChIP-seq data.

5) It would be very interesting to choose several genes of interest (out of the genes represented in the figures) and check the Pol II CTD status across these genes. This will reveal that status of transcription and could possibly provide an explanation of why the mature mRNA does not correlate with the GRO-seq observations. This should also be compared to the Mdm2 ChIP(s) requested above to see if genes to which Mdm2 binds cause the Pol II CTD to have different phosphorylation patterns.

6) The expression of the newly identified genes, while some were of low abundance, were not confirmed at the protein level. Is there a functional readout of these RNAs?

7) The authors identify the primary response genes upon p53 activation. Is there a better way to leverage the valuable list of directly regulated genes to infer anything about the p53 activation/repression network? For instance, since they find that p53 does not directly down regulate many genes, can they see evidence of upregulation of known or putative repressors that might be responsible for down-regulation of genes identified as such in microarray studies?

---

## [Author Response]

*1) It is possible that Nutlin does increase p53 levels in a small percent of the cells by 1 hour and it is those few cells with high p53 that have the increased transcription of p53 targets. Can the authors rule this out? Have they performed immunofluorescence analysis to see whether there are some ’jackpot’ cells where p53 protein*
*is rapidly induced by Nutlin?*

We are grateful for this comment, which inspired us to include new experiments, including the requested immunofluorescence, investigating the mechanism of action of Nutlin at the very short time point used for GRO-seq. First of all, the immunofluorescence discards the notion of ‘jackpot’ cells: although ∼3% cells show high p53 staining at the 1 hour time point of Nutlin treatment, this number is not significantly different from that observed in control p53 +/+ cells (Figure 1—figure supplement 1).

The greater question behind this comment is: how could we observe such drastic transactivation of p53 target genes at 1 hour of Nutlin treatment without any noticeable increase in total p53 levels? Although it is well established that MDM2 represses p53 both by masking its transactivation domain and by targeting it for degradation, the contribution of each mechanism to the regulation of p53 target genes has been very difficult to tease out experimentally. This is not a trivial issue, because previous studies employing steady state mRNA measurements concluded that prolonged p53 activation and higher p53 levels were differentially required for activation of apoptotic genes (5; 8; 11; 40; 50). However, GRO-seq demonstrates that a 1 hour time point of Nutlin treatment induces transcription of genes in every major pathway downstream of p53 (Figure 1). The observation that key survival (e.g. p21) and apoptotic genes (e.g. PIG3) show >6-fold increase in transcription at a time point preceding a proportional increase in total p53 levels as measured by Western blot, indicates that the mere unmasking of the p53 transactivation domain suffices to activate a multifaceted transcriptional program.

To further test this notion, we performed flow cytometry analyses using a monoclonal antibody (DO-1) that recognizes an epitope in the p53 N-terminal transactivation domain 1 (TAD1) that overlaps with the MDM2-binding surface competed by Nutlin (32). In fact, DO-1 is able to compete the p53-MDM2 interaction in vitro (6), in analogous fashion to Nutlin. Under the denaturing conditions of a Western Blot assay, where the p53-MDM2 complexes are fully disrupted, this antibody reveals no significant increase in total p53 levels at 1 hour Nutlin treatment (Figure 1). However, we posited that DO-1 reactivity should increase significantly upon Nutlin treatment under the fixed conditions employed in flow cytometry. Flow cytometry quantitation shows that, even before Nutlin treatment, p53 +/+ cells have significantly more DO-1 reactivity than p53 -/- cells (Figure 1). The functional importance of this ‘basal p53 activity’ is investigated at length later in our report (Figure 3). Interestingly, the p53 +/+ cell population shifts to significantly higher DO-1 reactivity at the 1 hour time point, as expected from epitope unmasking. A further increase is observed at 12 hours of Nutlin treatment, when total p53 levels have risen considerably as measured by Western blots (Figure 1).

Altogether, these results indicate that the low levels of p53 present in proliferating cancer cells suffice to directly activate a multifunctional transcriptional program, including many canonical apoptotic genes, upon unmasking of the p53 transactivation domain by Nutlin. However, as discussed later in the paper (Figure 4), this conclusion does not necessarily conflict with previous reports showing differential timing of mRNA induction between arrest and apoptotic genes as seen by steady state RNA measurements.

*2) While the Nutlin treatment allows direct ability to assay p53 function, a second method of p53 induction would greatly enhance these findings. While not direct necessarily, does an oncogenic or DNA-damage insult induce some of these new target genes? (even if a few were assayed by qPCR on a gene by gene basis). This would also address the survival vs. apoptotic regulation, since Nutlin treatment may have bias towards survival*.

We are grateful for this comment and the revised manuscript includes Q-RT-PCR for 14 of the novel genes discovered by GRO-seq showing that 12 of them are also induced by doxorubicin, a well established genotoxic p53 activating agent. This is now shown in Figure 1—figure supplement 1.

*3) The authors should perform an Mdm2 ChIP at a few select targets such as p21 to see whether the Mdm2 signal is down by one hour. This is an important experiment because it would extend and confirm the most interesting aspect of this study. Endogenously expressed Mdm2 ChIPs are not very commonly reported in the literature, but have been accomplished. See for example papers by J. Bargonetti and colleagues (White et al., Cancer Research 2006; Arva et al., JBC 2005)*.

*4) Relatedly, in the Results section, the authors conclude that p53 acts as a repressor at a subset of genes under basal conditions (also shown in supplemental*
Figure 3*). This is based on the comparison of expression in the p53 wt vs. null cells. Even though these genes are considered 'direct' targets from GRO-seq analysis after nutlin treatment, it is unclear whether they have evidence of p53 binding this small subset of genes, and since this is not an immediate knockdown experiment, shouldn't the authors consider secondary effects as they did in the first part of the paper. To support a direct role for p53 in repressing these promoters, the authors could ask whether p53 and/or MDM2 are at these basally regulated promoters by performing ChIP-qPCR or by using published ChIP-seq data*.

We thank the reviewers for these comment and we now include MDM2 ChIP data and p53 ChIP data for CDKN1A/p21 (basally activated) and two p53 target genes that are ‘basally repressed’ by low levels of p53 (*PTP4A1* and *HES2*) (Figure 3 and Figure 3—figure supplement 1). The MDM2 ChIPs were technically challenging in HCT116 cells as MDM2 does not contact DNA directly and we could not confidently define MDM2 chromatin binding above background levels. However, we successfully managed to obtain MDM2 ChIP signals that are significantly enriched over IgG controls by employing the SJSA cell line, which expresses high MDM2 levels due to MDM2 gene amplification. We also made slight modifications to our ChIP protocol washing conditions. These technical modifications are explained in the revised Methods section.

These results are now explained in the revised text as follows:

“Overall, these results indicate that p53 acts as a repressor at a subset of its targets in a manner that is relieved by Nutlin, suggesting that MDM2 recruitment by basal levels of p53 may repress transcription at specific loci. To test this hypothesis, we performed ChIP experiments for p53 and MDM2 under conditions matching the GRO-seq experiment. […] Of note, although Nutlin disrupts the interaction between the p53 N-terminus and the hydrophobic pocket in the N-terminal domain of MDM2, a second molecular interaction occurs between the p53 C-terminus and the MDM2 N-terminus that is not competed by Nutlin in vitro (34), which may explain why the MDM2 signal is not completely abrogated upon a short time point of Nutlin treatment. Western blots demonstrating specific MDM2 immunoprecipitation under the ChIP conditions utilized are shown in Figure 3—figure supplement 1.”

*5) It would be very interesting to choose several genes of interest (out of the genes represented in the figures) and check the Pol II CTD status across these genes. This will reveal that status of transcription and could possibly provide an explanation of why the mature mRNA does not correlate with the GRO-seq observations. This should also be compared to the Mdm2 ChIP(s) requested above ip to see if genes to which Mdm2 binds cause the Pol II CTD to have different phosphorylation patterns*.

As requested, we performed ChIP assays for Serine 5- and Serine 2-phosphorylated RNAPII matching our GRO-seq conditions. We now show ChIPs for *GDF15* (a ‘basally activated’ gene) and *PTP4A1* (a ‘basally repressed’ gene). Briefly, the phospho-RNAPII ChIPs reproduce the pattern of RNAPII behavior revealed by GRO-seq. The ‘basally repressed’ genes have lower RNAPII ChIP signals (and GRO-seq signals), when MDM2 is bound to the corresponding p53REs as shown by ChIP.

On a related note, we have now expanded our explanation of why GRO-seq and microarray measurements produce different pictures of p53 action, including a more detailed explanation of how microarrays fail to detect low abundance mRNAs.

*6. The expression of the newly identified genes, while some were of low abundance, were not confirmed at the protein level. Is there*
*a functional readout of these RNAs?*

We now provide Western blot data showing induction at the protein level for three novel p53 target genes: GRIN2C, PTCHD4 and RINL. These data are in Figure 1—figure supplement 1. Furthermore, we have expanded our explanation of why GRO-seq enabled us to discover so many novel genes not predicted by ChIP-seq/microarray. More specifically, we have included new p53 ChIP data for six of these novel genes (Figure 2—figure supplement 2), which shows that some of them have very distal p53 binding sites (i.e., >30kb from TSS), which would explain why they have been missed by ChIP-seq analysis requiring closer proximity to the TSS.

*7) The authors identify the primary response genes upon p53 activation. Is there a better way to leverage the valuable list of directly regulated genes to infer anything about the p53 activation/repression network? For instance, since they find that p53 does not directly down regulate many genes, can they see evidence of upregulation*
*of known or putative repressors that might be responsible for downregulation of genes identified as such in microarray studies?*

As explained in our manuscript, Ingenuity Pathway Analysis of the ‘repressed’ genes identified E2F4, RB and CDKN1A as the top regulators of this gene set. Since CDKN1A (p21) is the most actively transcribed p53 target gene according to GRO-seq, its strong upregulation upon Nutlin treatment would lead to dephosphorylation of Rb and transcriptional repression by E2F4 complexes, thus explaining a large fraction of the ‘repression’ observed. In the manuscript we also show that another large fraction of the repression can be explained by upregulation of miR-34a, a p53-inducible miRNA detected by GRO-seq as a primary target. Analysis of our ‘repressed’ gene set led to the observation that up to 72% of them are downregulated in this same cell type upon miR-34a overexpression. Therefore, two direct targets or p53 transactivation (CDKN1A and miR-34a) can drive most of the indirect gene repression seen at a late time point by microarrays.

We searched for other transcriptional repressors within our list of immediate direct p53 targets and identified PRDM1 (also known as BLIMP1). PRDM1 is as a DNA binding protein involved in gene repression that was previously characterized as p53 target (49). However, IPA analysis failed to identify PRDM1 as a top regulator of our ‘repressed’ gene set. In any case, it would be interesting to test in the future whether PRDM1 contributes to indirect gene repression downstream of p53. The revised manuscript mentions PRDM1 as another possible mediator of indirect gene repression downstream of p53.

Finally, our list of immediate p53 targets contains four other miRNAs beyond miR-34a (miR- 1204, miR-3189, miR4679-1 and miR-4692), which could also contribute to indirect post- transcriptional repression. Unfortunately, there is no experimental genomics data on the targets of these four miRNAs (only bioinformatic target predictions). The revised manuscript mentions these miRNAs and the possibility that they may also contribute to indirect gene repression downstream of p53.